# Hydrogel dressing integrating FAK inhibition and ROS scavenging for mechano-chemical treatment of atopic dermatitis

Yuanbo Jia[1,2,4], Jiahui Hu[2,3,4], Keli An[1,2], Qiang Zhao[2,3], Yang Dang[2,3], Hao Liu[1,2], Zhao Wei[1,2], Songmei Geng[3] ✉ & Feng Xu [1,2] ✉

Atopic dermatitis (AD) is a chronic skin disease caused by skin immune dys-homeostasis and accompanied by severe pruritus. Although oxidative stress and mechanical scratching can aggravate AD inflammation, treatment targeting scratching is often overlooked, and the efficiency of mechano-chemically synergistic therapy remains unclear. Here, we find that enhanced phosphorylation of focal adhesion kinase (FAK) is associated with scratch-exacerbated AD. We then develop a multifunctional hydrogel dressing that integrates oxidative stress modulation with FAK inhibition to synergistically treat AD. We show that the adhesive, self-healing and antimicrobial hydrogel is suitable for the unique scratching and bacterial environment of AD skin. We demonstrate that it can scavenge intracellular reactive oxygen species and reduce mechanically induced intercellular junction deficiency and inflammation. Furthermore, in mouse AD models with controlled scratching, we find that the hydrogel alleviates AD symptoms, rebuilds the skin barrier, and inhibits inflammation. These results suggest that the hydrogel integrating reactive oxygen species scavenging and FAK inhibition could serve as a promising skin dressing for synergistic AD treatment.

Atopic dermatitis (AD) is a chronic, recurrent, inflammatory skin disorder, often accompanied by severe pruritus, which severely affects the life quality of patients. AD has a growing prevalence of over 10% in the population, resulting in a huge global medical burden[1–3]. As an autoimmune disease with a complex etiology, AD cannot be cured with the currently available medical treatments. Most treatments focus on alleviating the symptoms of acute inflammation with steroid drugs and antihistamines, but continuous administration usually involves side effects (e.g., hyperglycemia, cushing syndrome, sleep disturbance)[4]. Therefore, there is an urgent need for an alternative therapeutic regimen or daily management approach for AD.

Mechanical scratching is an important reason for the aggravation and persistence of AD skin inflammation. The uncontrollable scratching as caused by pruritus promotes the release of TSLP from keratinocytes, which regulates the differentiation of T cells into $T_H2$ type, producing a large amount of $T_H2$ cytokines (e.g., IL-4, IL-13, and IL-31)[5,6]. These factors cause immunoreactive pruritus and promote the neurosensitization of peripheral itch by type 2 cytokines, promoting the formation of the itch-scratch cycle, which further causes deficiency of skin barrier-related proteins (e.g., Filaggrin, E-cadherin, Occludin)[7], and recruit inflammatory cells and promote the colonization of local microorganisms (e.g., *Staphylococcus aureus*)[8]. Many mechanosensors (e.g., TRP protein family, Piezo, integrins) are involved in the mechanotransduction process that promotes AD progression[9–13]. Although some antagonists of the TRP family can reduce inflammation in AD, their effects are limited[14]. One possible reason is that mechanical scratching involves multiple membrane mechanosensors, and blocking individual mechanosensors may be insufficient to shield the effects

---

[1]The Key Laboratory of Biomedical Information Engineering of Ministry of Education, Xi'an Jiaotong University School of Life Science and Technology, 710049 Xi'an, China. [2]Bioinspired Engineering and Biomechanics Center (BEBC), Xi'an Jiaotong University, 710049 Xi'an, China. [3]Department of Dermatology, The Second Affiliated Hospital, Xi'an Jiaotong University, 710004 Xi'an, Shaanxi, P. R. China. [4]The authors contributed equally: Yuanbo Jia, Jiahui Hu. ✉e-mail: gsm312@yahoo.com; fengxu@mail.xjtu.edu.cn

of mechanical stimulation. Numerous studies indicate that FAK acts as the convergence point of various mechanotransduction pathways that are closely related to inflammatory skin disorders[15–18]. Furthermore, FAK can also translocate to the nucleus to regulate inflammatory gene expression programs, including chemokines and cytokines, which reprogram the composition of the extracellular matrix (ECM)[19]. Thus, FAK may be a transmitter of inflammatory and physical signals in AD[20]. Besides, previous work has shown that FAK inhibitors can reduce mast cell-induced allergic reactions[21], suggesting the potential therapeutic cues for FAK in AD.

Oxidative stress, as an important factor in many inflammatory diseases, is increasingly recognized to be associated with the development of AD[22]. Imbalance of reactive oxygen species (ROS) not only directly damages skin cell membranes and organelles but also activates the NF-κB pathway, which induces the expression of proinflammatory cytokines (e.g., IL-4, IL-13, IL-33). These cytokines, in turn, enhance dermal inflammatory infiltration and histamine release in the affected skin, thereby developing or exacerbating AD by triggering pruritus or enhancing Th2 polarization[22,23]. Modulation of oxidative stress by ROS scavenging could be a promising approach to regulate the immune environment in AD[24]. However, the effects of oxidative stress and mechanical stimulation are highly coupled in AD. Vigorous scratching can enhance oxidative stress and lead to acute inflammation, while oxidative stress-mediated inflammation causes cellular mechanosensitization, which can enhance the inflammatory response of cells under scratching[25,26]. Therefore, single-factor treatment may not be sufficient to cope with the complex mechano-chemical coupling environment of AD, while the effectiveness of mechano-chemically synergistic therapy has not been explored yet.

Traditional AD care utilizes emollients or creams to maintain skin moisture or to deliver therapeutic drugs, which have a short epidermal barrier protection and drug residence time[27]. Hydrogels have been considered promising options for skin dressings due to their drug-release ability and diverse customized functions[28]. However, the unique pathology and mechanical environment of AD relative to other skin lesions place special requirements on dressings. First, AD occurs mostly in skin folds (e.g., armpits, neck), where the skin undergoes complex and extensive deformations. Therefore, the hydrogels should be soft enough with large stretchability. Next, involuntary scratching at AD skins may lead to local breakage of the dressing, which imposes requirements on the self-healing properties of the hydrogels. Finally, the hydrogels must have good tissue adhesion to ensure stable residence on the skin and avoid the use of additional tape or gauze. Although some previous works have used hydrogels as AD skin dressings to deliver therapeutic factors[29,30], these hydrogels often fail to meet the specific needs of AD management.

Herein, we developed an adhesive, stretchable, and self-healing hydrogel dressing based on borate ester[31], embedding polydopamine nanoparticles (PDA NPs) for ROS scavenging and liposome-encapsulated hydrophobic focal adhesion kinase inhibitor (FAKi-lipo) for FAK inhibition to synergistically treat AD (Fig. 1)[32]. PDA NPs have been widely used as ROS scavengers because of their numerous reducing catechol groups, while liposome encapsulation has been used to promote the drug loading and sustained release of hydrophobic drugs[33,34]. We validated the ROS scavenging efficiency of PDA NPs and the protective effect of FAKi-lipoLA on cells in response to mechanical stimulation in vitro. We further demonstrated the effect of mechano-chemically synergistic therapy of the hydrogel patch in a mouse AD model with controlled mechanical scratching.

## Results and discussion

### FAK phosphorylation is increased in AD skins and enhanced by scratching

To assess the relationship between FAK phosphorylation and AD, we performed hematoxylin-eosin (HE) staining of skin sections from human AD patients. Compared to normal skin, pFAK levels are significantly higher in the skin of AD patients and are concentrated in the epidermis that is directly exposed to mechanical stimulation from scratching (Fig. 2a–d and Supplementary Fig. 1). To further verify the effect of scratching on pFAK in AD, we constructed a mouse model of AD with controlled mechanical stimulation (Fig. 2e). Specifically, 45 µM Calcipotriene (MC 903) was used to induce initial AD immune imbalance and scratching was simulated by tearing the tape on the skin surface[35]. To avoid uncontrolled mechanical stimulation caused by mouse scratching, the injury skin was covered with gauze during the experiment. Compared to the group induced with MC 903 alone (AD group), mechanical scratching (AD_{scratch} group) significantly exacerbates AD development, as manifested by the increased clinical dermatitis score and epidermal thickness (Fig. 2a, b). However, mast cell infiltration is not significantly different in the two groups (Supplementary Fig. 2a, b). We further examined the expression of common inflammatory factors associated with AD (e.g., IgE, IL-4, TSLP, CCL-20, IL-13). Different from IgE and IL-4, the protein levels of TSLP, CCL-20, and IL-13 are significantly elevated in the scratched tissues (Supplementary Fig. 2c–g), suggesting that mechanical scratching may induce acute inflammatory responses through these factors, which is consistent with previous reports[36]. Disruption of the epidermal barrier is one of the hallmarks of AD lesions and will increase skin sensitization to allergens. Thus, we evaluated two important skin barrier proteins, i.e., E-cadherin (mainly expressed in the epidermis) and filaggrin (mainly expressed in the stratum corneum). Immunofluorescent staining of E-cadherin shows obvious epidermal thickening in the AD and AD_{scratch} groups. Unscratched AD skin shows the high fluorescence intensity of E-cadherin throughout the epidermis, while the fluorescence intensity of E-cadherin is significantly stratified in the AD_{scratch} group (Supplementary Fig. 2h, i). The filaggrin expression in the stratum corneum is also higher in the AD group than in the AD_{scratch} group (Supplementary Fig. 2j–l). Besides, compared to healthy skin, pFAK is enriched in the AD group and significantly elevated in the epidermis of the AD_{scratch} group (Fig. 2d). These results indicate that scratching may cause damage to epidermis barriers, and pFAK is associated with AD and significantly enhanced by mechanical scratching.

### Preparation and characterization of PDA NPs and FAKi-lipoLA

To scavenge ROS, we synthesized PDA NPs with high yield via a classical Stöber method[32]. We characterized the particle morphology using transmission electron microscopy (TEM) and Malvern Zetasizer ZS. PDA NPs have a monodispersed spherical structure (Fig. 3a), with an average diameter of $106.30 \pm 24.62$ nm (Fig. 3b), a hydrodynamic diameter of $242.20 \pm 4.01$ nm (Supplementary Fig. 3a), and good homogeneity with a polydispersity index (PDI) of 0.034 (Supplementary Fig. 3b).

Lauric acid-loaded liposomes (lipoLA) have shown potent antimicrobial activity and have been widely used in the treatment of skin diseases[37]. Thus, we chose lipoLA as the carrier for the delivery of drugs targeting exogenous mechanical stimulation of the AD skin. FAK is at the point of biological convergence of mechanotransduction and many inflammatory signaling pathways[16–18]. As the inhibitor of FAK, defactinib (VS-6063) was selected as the model drug and encapsulated in the lipoLA by the vesicle extrusion technique[38], generating monodisperse formulations (Fig. 3a). More specifically, we chose liposomes with a diameter close to 100 nm, considering the better skin penetration than larger liposomes[39,40]. Either lipoLA or FAKi-lipoLA holds a monodispersed spherical structure with an average diameter near 100 nm ($85.69 \pm 13.90$ nm for lipoLA, $81.70 \pm 13.54$ nm for FAKi-lipoLA), as observed from transmission electron microscopy (TEM) and Malvern Zetasizer ZS (Fig. 3b). In addition, liposomes prepared by the vesicle extrusion technique have a monotonic size distribution with a particular dispersity index (PDI) less than 0.2 (Supplementary Fig. 3b). Moreover, lipoLA and FAKi-lipoLA have the zeta potential about

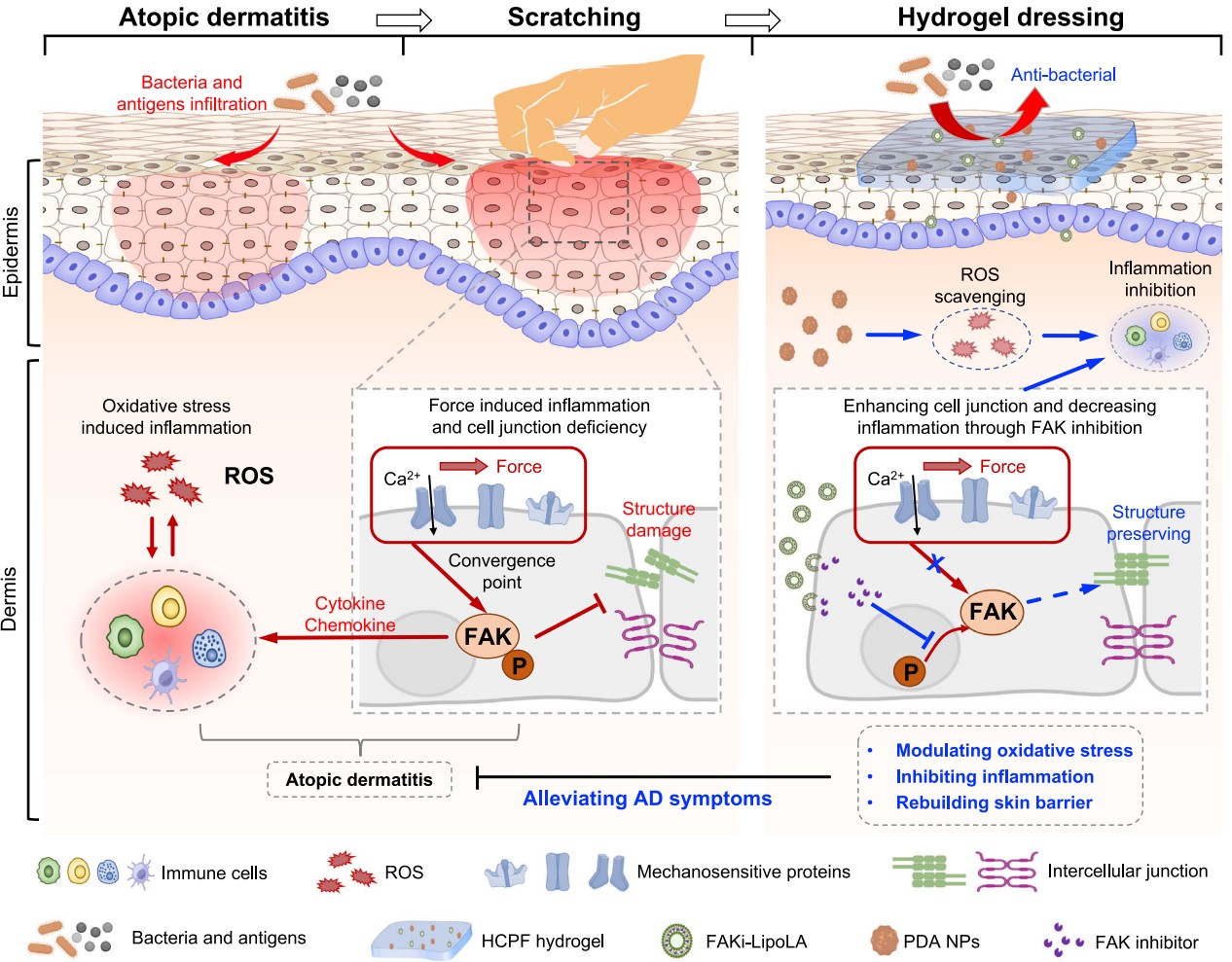

**Fig. 1 | Schematic of HCPF hydrogel for AD treatment.** The left schematic shows the inflammatory response of AD due to oxidative stress and the exacerbating effect of mechanical scratching on AD. The right schematic depicts the hydrogel dressing synergistically treating AD by scavenging reactive oxygen species with inhibition of FAK phosphorylation.

−51.53 ± 7.05 mW and −68.87 ± 4.19 mW, respectively (Fig. 3c). These results indicate that all the NPs have uniform size and great colloidal stability, making them suitable for drug delivery for the treatment of AD.

### The adhesive HCPF hydrogels exhibit good tissue compliance, self-healing, and antimicrobial properties

Dopamine (DA) modified hyaluronic acid (HA-DA) and phenylboronic acid (PBA) modified carboxymethyl chitosan (CMCS-PBA) were selected as the backbone of hydrogels (Supplementary Fig. 4a, b). PDA NPs and FAKi-lipoLA were separately or co-introduced into the hydrogel networks (Fig. 3d), termed as HC (blank hydrogels), HCP (HC hydrogels with PDA NPs), HCF (HC hydrogels with FAKi-lipoLA), and HCPF (HC hydrogels with PDA NPs and FAKi-lipoLA).

For chronic AD skin management, the dressing must have sufficient tissue adhesion to ensure sustained skin retention and drug delivery during daily activities while it can be easily removed to avoid secondary damage to mechanically sensitive AD skin. Catechol groups of dopamine and PDA have been widely incorporated into different materials to enhance tissue adhesion[41]. Therefore, we speculated that HC hydrogels containing HA-DA can achieve effective adhesion to the skin surface, and the introduction of PDA NPs can enhance such adhesion[42]. To verify this, we carried out shear tests using pig skin. Compared with HC hydrogels (12.56 ± 2.09 kPa), the introduction of FAKi-lipoLA does not change the adhesion strength of the hydrogels

on the surface of pig skin (12.63 ± 2.02 kPa), while the adhesion strength is improved after adding 1 mg/ml PDA (19.71 ± 3.34 kPa for HCP hydrogels and 20.05 ± 1.27 kPa for HCPF hydrogels) (Fig. 3e). Tests on human skin also showed that the HCPF hydrogels can adhere to tissue surfaces and withstand various skin deformation or joint movement and can be removed without residue (Fig. 3h).

Ideal skin dressings for AD skin management should be soft with high stretchability. Thus, we characterized the mechanical properties of the hydrogel by rheological and tensile experiments. A frequency sweep of the shearing test by a rheometer revealed that the HC hydrogels have remarkable viscoelastic properties with a storage modulus of 1.24 ± 0.17 kPa, which is smaller than the modulus of human skin[43]. Loading with PDA NPs and FAKi-lipoLA has no significant effect on the hydrogel modulus (Fig. 3f and Supplementary Fig. 4c). Tensile experiments demonstrate that the HCPF hydrogels have high stretchability (over 800%) (Fig. 3h), which would enable the adaptation to different skin deformations.

For AD skin that is often subjected to mechanical scratching, self-healing of the hydrogels is critical to maintaining complete coverage of the dressing on the skin surface after scratching. To evaluate the self-healing efficiency, we performed time-related strain sweep tests by applying a large shear strain (300%) mimicking mechanical scratching to destroy the HCPF hydrogels. The hydrogel was disrupted at 300% strain (the storage modulus G′ decreases significantly to less than the loss modulus G″). When the strain is recovered to 1%, G′ recovers

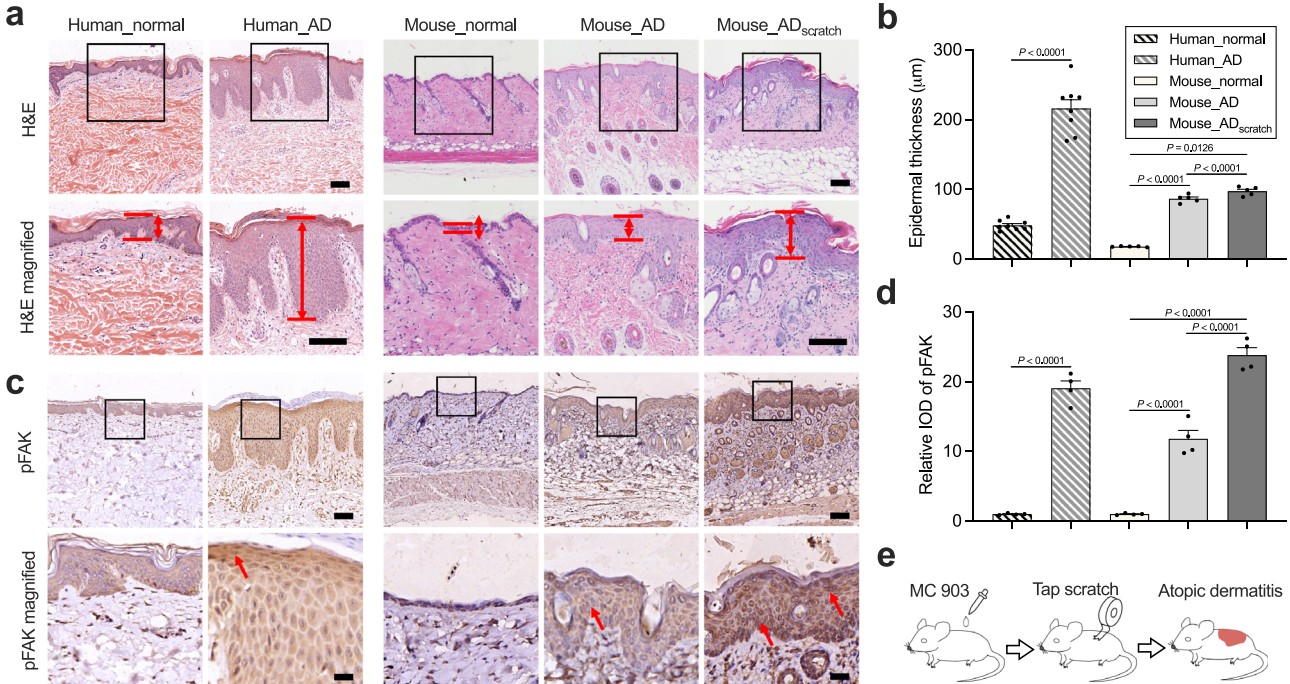

**Fig. 2 | pFAK is upregulated in AD skin, especially after scratching.**
**a** Representative hematoxylin and eosin (H&E) staining of skin section of humans and mice. The black boxed area is enlarged below. The space between red lines denotes the epidermal thickness. Scale bar, 100 μm. **b** Epidermal thickness quantified from H&E staining in (**a**), For human skin, $n = 8$. For mouse skin, $n = 5$. **c** Immunohistochemistry staining of pFAK in each group. Scale bar, 100 μm for the images above and 25 μm for the magnified images. The red arrows indicate the area of high DAB staining (high pFAK expression). **d** Relative IOD of pFAK quantified from images of (**c**), $n = 4$ mice. **e** Schematic diagram of scratching-controlled AD models. In (**b**) and (**d**), all data are shown as mean ± s.e.m. For the human sample, data are compared by a two-tailed Student's $t$-test. For mice sample, data are compared by one-way ANOVA followed by Bonferroni's post hoc test.

rapidly within 20 s, indicating that the disrupted structure is restored (Fig. 3g). Adhesion experiments on the skin surface also showed that the dissected HCPF hydrogels could quickly heal within 10 s and withstand the large deformation at the joint (Fig. 3h). The good self-healing ability of HCPF hydrogels benefits from the fast dissociation-reconstruction kinetics of boronate ester bonds, i.e., the boronate ester bonds dissociate under large strain and undergo rapid addition reactions after the load is removed[44]. These results suggest that the tissue-adhesive, soft, highly stretchable, and self-healing HCPF hydrogels hold great potential as ideal dressings for AD skin.

AD skin is often accompanied by *Staphylococcus aureus*, and scratching increases the risk of bacterial infection[8]. Using dressings to control bacterial growth may benefit AD and reduce the risk of complications. Carboxymethyl chitosan, the main component of HCPF hydrogel, has been reported to have good antibacterial properties[45]. We examined the antibacterial properties of different hydrogels using a classical plate coating method. Relative to the control group, the number of colonies of bacteria co-incubated with the hydrogel is significantly reduced (Fig. 3i). The absorbance of the bacterial suspension after hydrogel treatment also revealed more than 98% clearance efficiency of each group of hydrogels against *S. aureus* (Fig. 3j). These results indicate that HCPF hydrogels could effectively kill bacteria to reduce the chance of bacterial infection in AD.

**PDA NPs, FAKi-lipoLA, and HCPF hydrogels are cytocompatible**
To verify the cytocompatibility of each component to optimize the drug loading, we evaluated the cytotoxicity of different concentrations of PDA NPs, FAKi-lipoLA, and HCPF hydrogels on HaCaT cells using MMT assay. PDA NPs have no significant effect on cell viability at concentrations below 1 mg/ml. When the concentration is increased to 2 mg/ml, there is a slight decrease in cell viability (Supplementary Fig. 5a). Similarly, FAKi-lipoLA up to 1 mg/ml has no significant effect

on cell viability (Supplementary Fig. 5b). Further, we verified the biocompatibility of the drug-loaded hydrogel. Compared to the control group, HCPF hydrogel loaded with 1 mg/ml PDA NPs and 1 mg/ml FAKi-lipoLA has no significant effect on cell viability (Supplementary Fig. 5c), which is also demonstrated by co-staining of Calcein-AM and propidium iodide (PI) (Supplementary Fig. 5d). Hence, these results indicate that the HCPF hydrogel loaded with PDA NPs and FAKi-lipoLA has good biocompatibility and should be safe as skin dressings.

**PDA NPs and HCPF hydrogels can scavenge ROS in vitro**
To assess the ROS scavenging capability of PDA NPs and HCPF hydrogels, we first evaluated the in vitro scavenging efficiency of PDA NPs against two major free radicals in vivo, i.e., hydroxyl radicals (HO·) and superoxide radicals ($O_2^{·-}$). We observed that the fluorescence intensity of 2-hydroxyterephthalic acid decreases with increasing concentration of PDA NPs (Fig. 4a), indicating that PDA NPs have concentration-dependent scavenging of HO· produced by 10 mM $H_2O_2$. When the concentration of PDA NPs reaches 150 μg/ml, no fluorescence signal could be detected, indicating the complete clearance of HO·. Besides, we assessed the $O_2^{·-}$ scavenging efficiency of PDA NPs using nitro blue tetrazolium (NBT) assay. The inhibition ratio of photoreduction of NBT as reflected by absorbance at 560 nm showed that the scavenging efficiency of PDA NPs for $O_2^{·-}$ reaches a peak at the concentration of 75 μg/ml (-82%), after which the increase in concentration has no significant effect on the improvement of clearing efficiency (Fig. 4b). Due to the controlled-release effect of the hydrogel, the PDA NPs embedded in the hydrogel network cannot disperse rapidly in the surrounding tissues. Therefore, we further evaluated the total antioxidant capacity of the HCPF hydrogels. HCPF hydrogels exhibit increased total antioxidant efficiency with increasing concentrations of PDA NPs (Supplementary Fig. 5e). Further, we investigated the cytoprotection under hyper oxidative conditions and

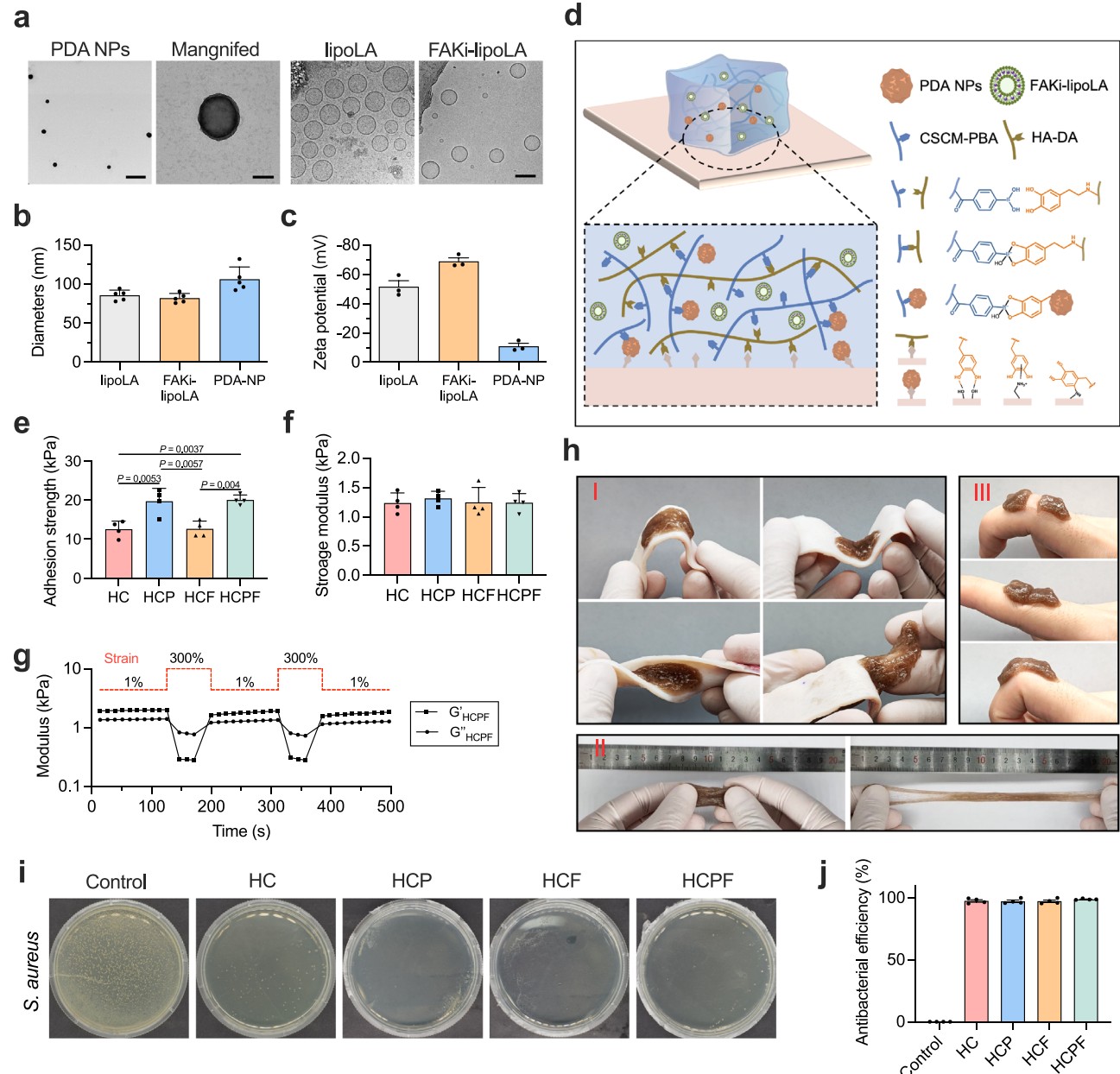

**Fig. 3 | Design and characterization of PDA NPs, FAKi-lipoLA, and HCPF hydrogel. a** TEM images of PDA NPs and FAKi-LipoLA. Scale bar, 1 μm and 100 nm for the magnified field. $n = 5$ samples with similar results. **b** Particle diameters calculated from TEM images in (**a**). $n = 5$ samples. Data are shown as mean ± s.d. **c** Zeta potential of different particles. $n = 3$ samples. Data are shown as mean ± s.d. **d** Schematic illustration of crosslinking mechanism and adhesive mechanism of hydrogels. **e** Adhesion strength of different hydrogels on pig skin, $n = 4$ samples. Data are shown as mean ± s.d. **f** Storage modulus of different hydrogels, $n = 4$ samples, and no significant difference was found among the four groups. Data are shown as mean ± s.d. **g** Self-healing property of HCPF hydrogel tested by rheology tests. **h** Demonstration of hydrogel properties. (I) Tissue adhesion on pig skin; (II) stretchability of HCPF hydrogels; (III) self-healing of HCPF hydrogels. **i** Images of survival *S. aureus* bacteria clones. **j** Antibacterial efficiency of different hydrogels calculated by the absorbance of the bacterial solution, $n = 4$ tests. Data are shown as mean ± s.e.m. In (**e**), (**f**), and (**j**), data are compared by one-way ANOVA followed by Bonferroni's post hoc test.

intracellular ROS scavenging ability of HCPF hydrogels. We co-incubated HCPF hydrogels with $H_2O_2$-treated HaCaT cells. First, 1 mM $H_2O_2$ was used to simulate a highly cytotoxic oxidative environment. MTT assay revealed that HCP hydrogels loaded with over 0.8 mg/ml PDA NPs show effective protection of HaCaT cells (>80% cell viability) compared with the $AD_{scratch}$ group (about 33% cell viability) (Fig. 4c). Next, to verify the intracellular ROS scavenging ability of HCPF hydrogels, exogenous Rosup reagent was added to the culture medium to induce the production of intracellular ROS. The fluorescent probe showed that HCP hydrogels significantly inhibit intracellular ROS production (Fig. 4d, e). These results demonstrate the antioxidant ability of PDA NPs and the oxidative protection of HCPF hydrogels in cells.

## FAK inhibition decreases stretch-induced inflammation and E-cadherin deficiency in vitro

To assess the inhibitory efficiency of FAKi-lipoLA on intracellular FAK expression, we used different concentrations of FAKi-lipoLA to treat HaCaT cells. We observed from western blotting tests that FAKi-lipoLA with a concentration of 100 μg/ml significantly inhibits the expression of phosphorylated FAK (pFAK). When it comes to 200 μg/ml, the expression of FAK is significantly reduced, and pFAK is almost not

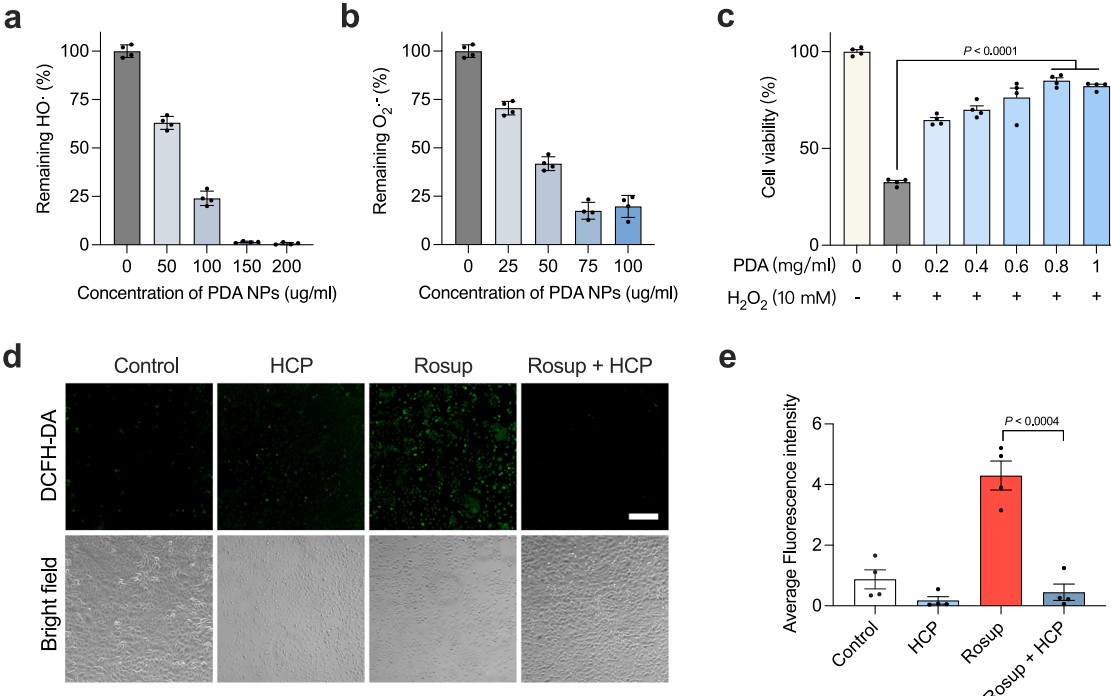

**Fig. 4 | PDA NPs and HCP hydrogels can efficiently scavenge ROS. a**, **b** HO· and O₂·⁻ scavenging efficiency of PDA NPs with different concentrations. $n = 4$ samples and data are shown as mean ± s.d. **c** Cytoprotective effect of HCP hydrogels with different concentrations of PDA NPs against 10 mM $H_2O_2$. $n = 4$ tests and data are shown as mean ± s.e.m. and compared by one-way ANOVA followed by Bonferroni's post hoc test. **d** Intracellular ROS scavenging of HCP hydrogels. Cells seeded were co-incubated with HCP hydrogels loaded with 1 mg/ml PDA NPs. $n = 4$ tests with similar results. Scale bar, 100 μm. **e** Fluorescence intensity of DCFH-DA reveals intracellular ROS level of HaCaT cells. $n = 4$ tests and data are shown as mean ± s.e.m. **** indicates $P < 0.0001$ compared by compared using a two-tailed Student's $t$-test.

expressed. Considering that pFAK is the actual executor in the cellular mechanical response, the results indicate that FAKi-lipoLA can effectively inhibit the biological activity of FAK (Fig. 5a). Further, we tested the release of FAKi-lipoLA encapsulated in the hydrogel. We used fluorescent nylon red to label liposomes and tested the fluorescence intensity of HCF hydrogel extracts in Dulbecco's modified Eagle medium (DMEM) medium at different times. We placed HCF hydrogels in a transwell to simulate the skin–hydrogel interface. The result showed that the FAKi-lipoLA in HCF could be released about 13% at 24 h and 18% at 72 h (Fig. 5b).

Scratching in AD patients often imposes localized stretches with large deformation on the skin, which leads to impairment of the epidermal barrier and acute inflammation. To simulate the mechanical scratching in vitro, we used a custom stretching device to apply different cyclic stretching to HaCaT cells. Cells of the HC group (blank control, cultured with HC hydrogel extract) and the HCF group (cultured with HCF hydrogel extract) were separately seeded in the two compartments of a PDMS membrane to ensure that the two groups of cells were subjected to the same stretching (Fig. 5c). E-cadherin is a major component of epidermal intercellular junctions, and its loss of expression correlates with impaired epidermal cell function and morphology[46]. Therefore, we examined the integrality of intercellular E-cadherin by immunofluorescence staining. Interestingly, little E-cadherin staining was observed in the unstretched cells. Considering that in vivo skin tissues are continuously subjected to stretching at low strain, and previous studies have reported that proper mechanical tension promotes intercellular E-cadherin formation[47,48]. We stretched HaCaT cells at 5% strain for 4 h. The results showed that clear and intact intercellular E-cadherin was formed in the HC and HCF groups. Therefore, we used cells with 5% stretching for 4 h to mimic the condition of normal skin. Based on this, we further stretched the cells at 30% strain for 4 h, which has been proven to cause mechanical damage

to cells[49]. After stretching, E-cadherin in the HC group showed a significant deficiency, while this could be rescued in the HCF group (Fig. 5d, e). Several studies have demonstrated that FAK inhibition decreases actin contractility[50]. To verify whether FAK inhibition rescues E-cadherin deficiency by decreasing cellular contractility, we treated cells with the non-muscle myosin II inhibitor Blebbistatin. Blebbistatin above 50 μM will lead to abnormal cell morphology and even reduced cell spreading, which will directly impair intercellular junctions. In contrast, 1000 μg/ml of FAK-lipoLA (the maximum loading concentration of FAK-lipoLA in hydrogels) did not affect the morphology of HaCaT cells (Supplementary Fig. 6a, b). Therefore, 10 and 30 μM were chosen as working concentrations. Inhibition of actomyosin contractility by Blebbistatin rescues stretch-induced E-cadherin deficiency (Supplementary Fig. 6c, d). However, compared to 30 μM, 10 μM Blebbistatin has a significantly reduced ability to rescue E-cadherin deficiency. Further, we test the secretion of inflammatory factors after stretch using enzyme-linked immunosorbent assay (ELISA). For the HC group, we observed that TNF-α, TSLP, and CCL-20 secretions are significantly increased after 30% stretching, while treatment with HCF extract significantly reduces their expressions under 30% stretching. Besides, for the 5% stretching groups, there is no difference between the HC and HCF groups, indicating that FAK-lipoLA does not affect the expression of inflammatory factors under 5% stretching (Fig. 5f). In conclusion, these results demonstrate that HCF can effectively release FAKi-lipoLA to reduce inflammation and intercellular junction deficiency in HaCaT cells under large mechanical stretching by inhibiting pFAK levels and reducing cellular contractility.

## HCPF hydrogels alleviate AD symptoms in a mouse model with controlled scratching

To evaluate the synergistically therapeutic potential of HCPF hydrogels, we used a mouse AD model with controlled mechanical

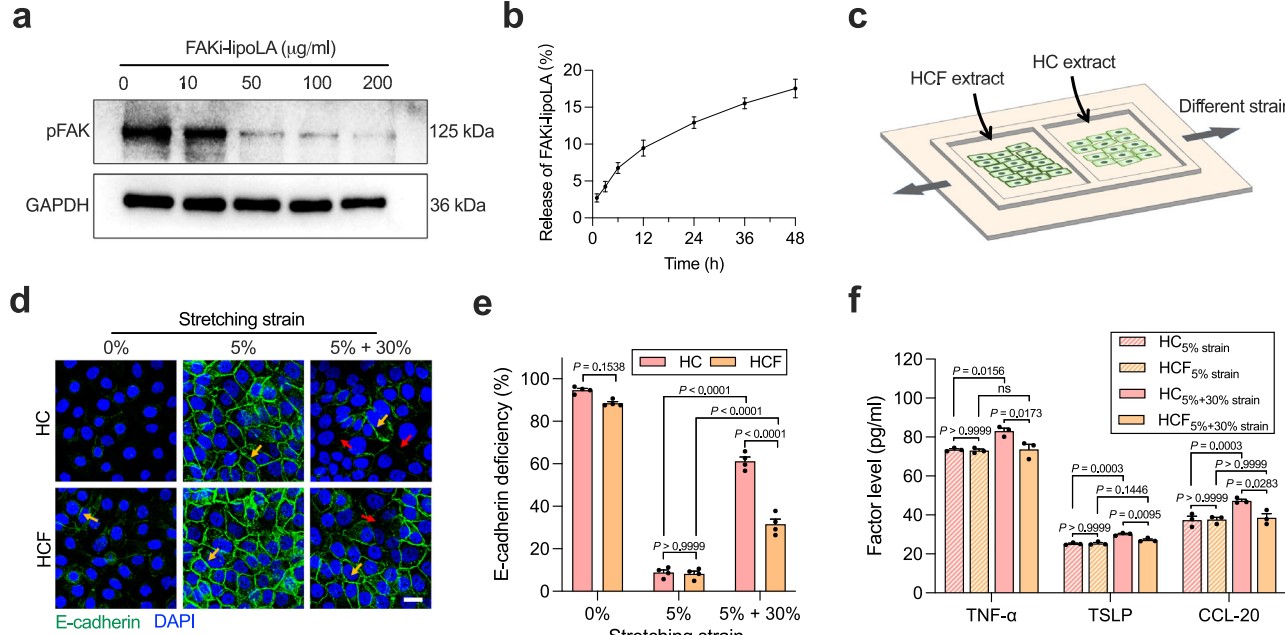

**Fig. 5 | Inhibition of FAK protects cells from mechanical damage in vitro.**
**a** Western blotting results of pFAK inhibition in HaCaT cells by FAKi-lipoLA. **b** Time-related FAK-lipoLA release from HCF hydrogels placed in a transwell. Data are shown as mean ± s.d. **c** Schematic diagram of cell stretching experiment. **d** Fluorescent staining of HaCaT cells after stretching with E-cadherin (green) and DAPI (blue). Scale bar, 50 μm. **e** E-cadherin deficiency of HaCaT cells calculated from staining images in (**d**). $n = 4$ tests. **f** TNF-α, TSLP, and CCL-20 level in the culture medium of HaCaT cells after stretch. $n = 3$ tests. In (**e**) and (**f**), data are shown as mean ± s.e.m. and compared by two-way ANOVA followed by Bonferroni's post hoc test.

stimulation as previously mentioned (Fig. 1e). Next, the mice were randomly divided into six groups (i.e., AD$_{scratch}$, HC hydrogels, HCP hydrogels, HCF hydrogels, and HCPF hydrogels) and treated with different hydrogels. During treatment, 25 μM MC 903 and mechanical stimulation were continuously applied to mimic the continuous allergen exposure and the scratching during treatment (Fig. 6a). After 10 days of treatment, pFAK expression in skin tissue is significantly decreased in the HCF and the HCPF groups, indicating that FAKi-lipoLA delivered via hydrogel could effectively regulate FAK phosphorylation in the skin (Supplementary Fig. 7). Dermatitis score indicated that the epidermal damage in the blank HC group (8.43 ± 0.37) is not significantly improved compared with the AD$_{scratch}$ group (9.29 ± 0.29), with obvious dandruff, skin damage, and edema (Fig. 6b, c). In comparison, both PDA NPs-loaded HCP hydrogels (6.86 ± 0.34) and FAKi-lipoLA-loaded HCF hydrogels (4.57 ± 0.30) significantly improve skin lesions. Compared to the other groups, the synergistically treated HCPF group has the lowest dermatitis score (3.14 ± 0.34), with only a small amount of dandruff and mild edema visible on the tissue. Similar to the dermatitis score, HCPF could significantly improve epidermal hypertrophy as reflected by the HE staining, which is the representative indicator of AD (Fig. 6d, e). Further, the HCPF group has minimal mast cell infiltration, as reflected by the toluidine blue staining (Fig. 6f, g). These results indicate that HCPF hydrogels could effectively improve the skin pathology of AD in the presence of scratching.

**HCPF hydrogels decrease in vivo inflammation, oxidative damage, and bacterial infection and repair the epidermal barrier of AD skin**
To explore the mechanisms by which HCPF hydrogels alleviate AD symptoms, we first assessed the inflammation of AD tissue in different groups using ELISA. The expressions of inflammatory factors CCL-20 and TSLP, which are associated with mechanical scratching, are significantly lower in the HCF and HCPF groups compared to the AD$_{scratch}$ group, while there is no significant difference in the HCP group (Fig. 7a, b), indicating that clearance of ROS may be insufficient to inhibit the

skin inflammation caused by mechanical scratching. The expression levels of IgE, IL-4, and IL-13 are significantly lower in the HCP, HCF, and HCPF groups (Fig. 7c–e and Supplementary Fig. 8a–e), suggesting that both ROS inhibition and FAK inhibition could alleviate AD symptoms by inhibiting the secretion of these inflammatory factors. Besides, to test the ROS scavenging effect of hydrogels in vivo, fluorescent staining was performed on slides to detect 8-hydroxy-2′-deoxyguanosine (8-OHdG) (Supplementary Fig. 8f), an oxidation product of deoxyguanosine residues in DNA[51,52]. The results showed that HCP and HCPF hydrogels loaded with PDA NPs could effectively reduce DNA damage in the epidermis and dermis (Supplementary Fig. 8g). The fluorescence intensity is slightly reduced in the HC and HCF groups relative to the AD$_{scratch}$ group, which may be due to the ROS scavenging effect of the dopamine groups in the HC hydrogels and the reduced acute inflammation by FAK inhibition.

We further examined the repair effect of different treatments on the epidermal barrier using fluorescence staining. We observed obvious epidermal thickening in the AD$_{scratch}$ group and obvious deficiency of the E-cadherin in the superficial layer, manifested as decreased fluorescence intensity and structural damage (Fig. 7f). We quantified the thickness of E-cadherin deficient tissue to assess the extent of epidermal barrier disruption. The results showed that, relative to the AD$_{scratch}$ group, all hydrogel-treated groups reduced the deficiency of E-cadherin (Fig. 7g). The HCP group has no improvement relative to the blank HC group, suggesting that scavenging of ROS alone may not be effective enough for scratching-induced epidermal barrier disturbance. However, HCF and HCPF hydrogels inhibiting FAK significantly reduce the deficiency of the E-cadherin, which is consistent with the results of cell stretching experiments (Fig. 5f). Besides, epidermal thickness as quantified by E-cadherin staining showed similar results to H&E staining (Supplementary Fig. 8h). Further, we checked the presence of filaggrin, an important structural molecule of the stratum corneum, using immunohistochemistry (IHC). We observed that filaggrin has the highest enrichment in the stratum corneum in the HCF and the HCPF groups (Fig. 7h, i), indicating that

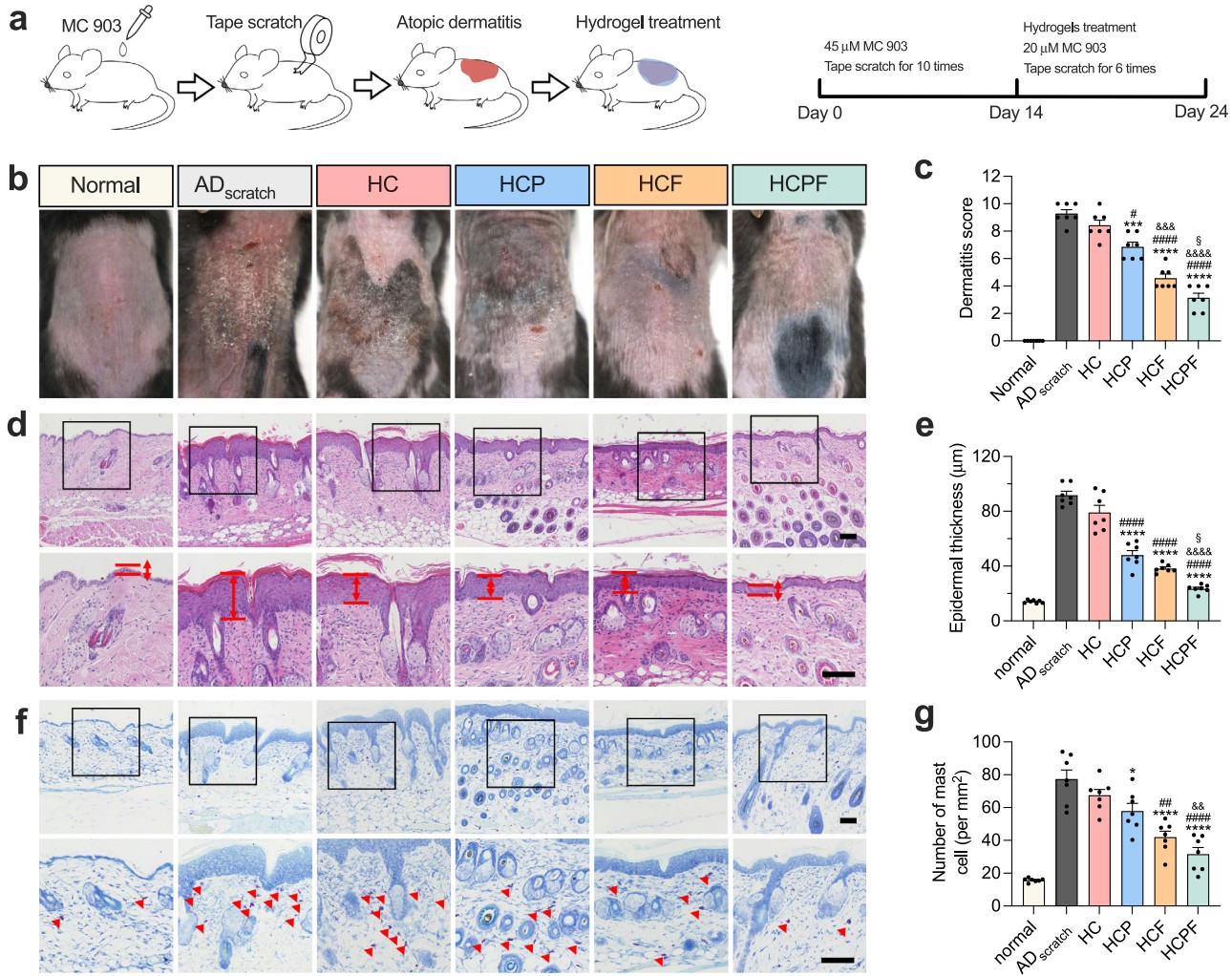

**Fig. 6 | HCPF hydrogels reduce lesions and mast cell infiltration in mice AD skins. a** Schematic diagram of animal experiments. **b** Representative photographs of the dorsal skin of each group on day 24. **c** Dermatitis score of each group assessed from photographs in (**b**). **d** Representative H&E staining of skin section. The black boxed area is enlarged below. The space between red lines denotes the epidermal thickness. Scale bar, 100 μm. **e** Epidermal thickness quantified from H&E staining. **f** Representative toluidine blue staining of skin section. The red triangle denotes dermal mast cells. Scale bar, 100 μm. **g** Measurement of the density of mast cells for each group after treatments. In (**c**–**g**), $n = 7$ mice, and all data are shown as mean ± s.e.m. *, #, &, and § indicate data compared with AD$_{scratch}$, HC, HCP, and HCF, respectively. *, **, ***, and **** indicate $P < 0.05$, $P < 0.01$, $P < 0.001$, and $P < 0.0001$ compared by one-way ANOVA followed by Bonferroni's post hoc test, respectively.

FAK inhibition can effectively reduce the disruption of the epidermal barrier as caused by mechanical scratching. Further, to test the in vivo antimicrobial efficiency of hydrogels, we further evaluated the therapeutic efficacy of HCPF hydrogel in the presence of *S. aureus* colonization. The results showed that HCPF hydrogel could effectively inhibit *S. aureus* in the skin of mice and effectively alleviate AD symptoms under bacterial infection conditions (Supplementary Fig. 9). Taken together, the HCPF hydrogels could effectively reduce the inflammation, oxidative damage, and *S. aureus* infection in AD tissues and the impairment of the epidermal barrier caused by mechanical scratching to synergistically alleviate AD symptoms.

## Discussion

Hydrogels have been widely used as wound dressings, but their applications in skin lesions with mechanical scratching (e.g., AD and psoriasis) are relatively rare, mainly because scratching can easily lead to dressing breakdown and failure. Here, we developed a rapid self-healing hydrogel based on the borate ester interaction between phenylboronic acid and catechol to ensure rapid reconstruction of the broken barrier while enhancing tissue adhesion and controlled release

of PDA NPs[53]. In addition to self-healing dressings, hydrogels with outstanding toughness that can resist scratching may be another promising approach[54]. Such a skin dressing may facilitate the dispersion of scratching stress on skin and reduce allergen infiltration. However, how to balance tissue compliance (i.e., flexibility) with scratching resistance (i.e., toughness) is an issue to be addressed for existing hydrogels. On the other hand, to distinguish the effects of ROS scavenging and FAK inhibition on AD and their synergistic therapeutic effects, we chose a dual-nanoparticles delivery system (PDA NPs and FAKi-LipoLA). Considering the extensive research of polydopamine in the construction of core-shell nanoparticles, delivery of PDA and FAKi can be integrated into future work.

Despite the increasing interest in mechano-chemically synergistic therapies for various diseases (e.g., cancers, myocardial infarction, wound healing), there has been little research on synergistic therapy for AD. As an immune disease, the inflammatory responses to biochemical cues and mechanical cues in AD skins are complex and coupled. For example, scratching-enhanced cytokine secretion can activate multiple receptors (e.g., IL-4Rα, IL-13Ra1, and IL31RA) on the surface of skin sensory neurons, which finally promotes the release of

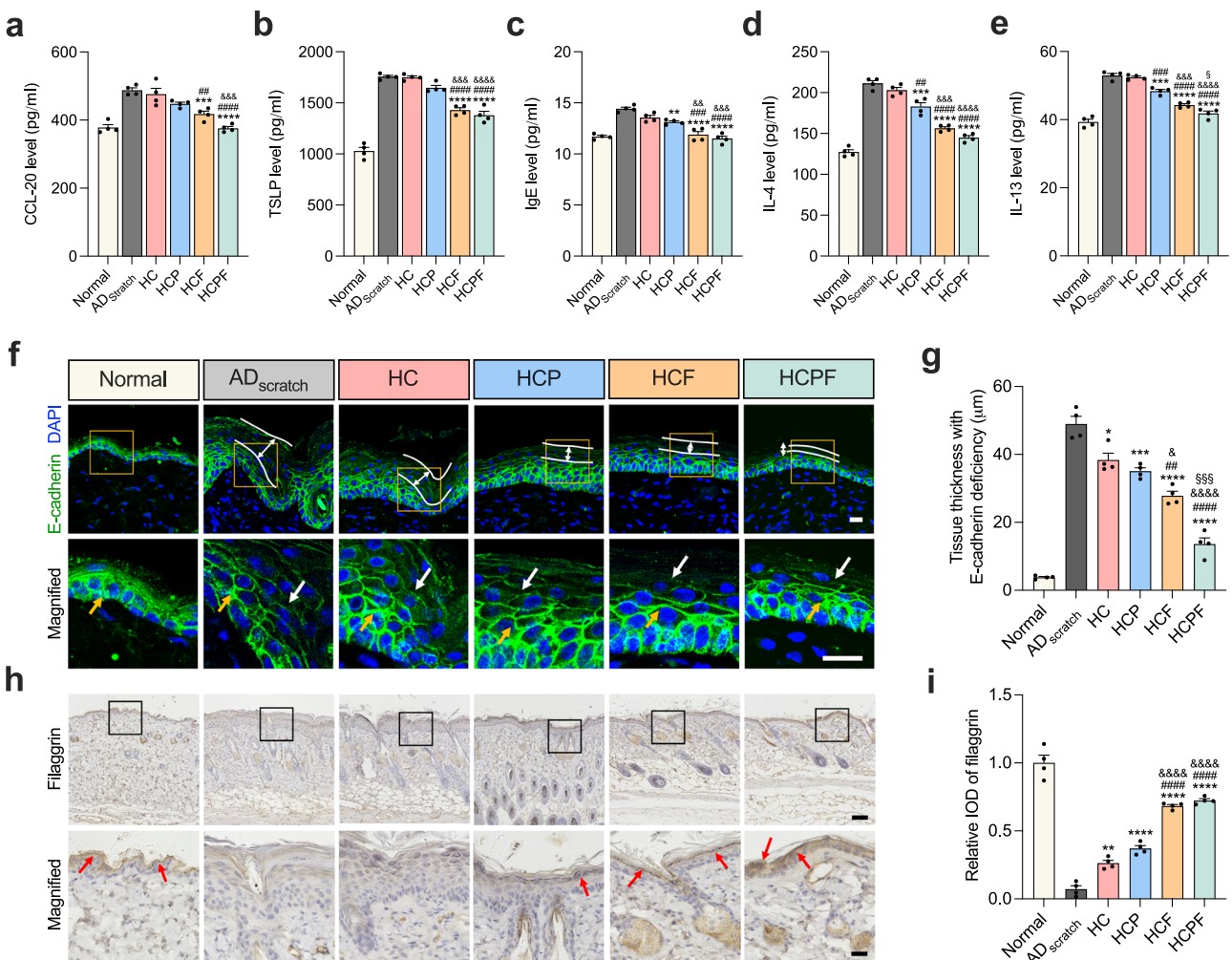

**Fig. 7 | HCPF hydrogels reduce inflammation and epidermal barrier damage in AD mice. a–e** CCL-20, TSLP, IgE, IL-4, and IL-13 levels in tissues of each group, $n = 5$ mice. **f** Fluorescent staining of E-cadherin (green) and DAPI (blue) in each group. Scale bar, 20 μm. The white lines denote the deficient area of E-cadherin in the epidermis. In the magnified images, the yellow arrows denote intact intercellular junctions represented by intact E-cadherin, while the white arrows indicate deficient E-cadherin. **g** Quantification of tissue thickness with E-cadherin deficiency from images in (**f**), $n = 4$ mice. **h** Immunohistochemistry staining of filaggrin in each group. Scale bar, 100 μm for the images above and 25 μm for the magnified images. The red arrows indicate the area of high DAB staining (high filaggrin expression). **i** Relative IOD of filaggrin quantified from images of (**h**), $n = 4$ mice. In (**a–e**), (**g**), and (**i**), all data are shown as mean ± s.e.m. *, #, &, and § indicate data compared with AD$_{scratch}$, HC, HCP, and HCF, respectively. *, **, ***, and **** indicate $P < 0.05$, $P < 0.01$, $P < 0.001$, and $P < 0.0001$ compared by one-way ANOVA followed by Bonferroni's post hoc test, respectively.

neuromediators (e.g., SP, CGRP, BNP, PACAP, and ET-1) from peripheral nerve endings, thereby exacerbating inflammation[55]. Meanwhile, inflammation has been proven to enhance TRPA1-mediated mechanical sensitization in pruritus receptors or nociceptors[56,57]. Despite being less studied in AD, the coupling and interaction between oxidative stress and mechanical stimulation have raised increasing attention. Mechanical stimulation has been widely known to promote ROS production through inflammatory factors such as TNF-α or direct mechanotransduction[58]. Meanwhile, oxidative stress-mediated inflammation can enhance the mechanosensitization of cells, which in turn amplifies the effects of mechanical stimulation[59]. For example, $H_2O_2$ enhances mechanically activated TRPA1-dependent $Ca^{2+}$ responses in the periodontal ligament, while free radical scavengers can inhibit nociceptive behaviors[26]. In this work, we found that scratching and oxidative stress have overlapped but are not identical inflammatory regulating mechanisms, while ROS scavenging and FAK inhibition improve AD symptoms in a synergistically enhanced manner. In particular, inflammation induced by scratching may cause serious consequences, as evidenced by the better therapeutic effect of FAK inhibition treatment (HCF group) than ROS clearance (HCP group)

(Fig. 6c) and failure of HCP treatment for CCL-20 and TSLP inhibition (Fig. 7a, b). These highlight the critical role of mechano-immunology in AD progression and the benefits of mechano-chemically synergistic treatment for AD. It should be mentioned that although treatment with hydrogels relieved AD symptoms, in the HCPF group, skin lesions (Fig. 6), inflammatory markers (such as TSLP and IL-4, Fig. 7b, d), and intercellular junctions (Fig. 7g, i) are still not restored to the level of healthy mice. One possible reason is that the mice were exposed to a little dose of MC 903 and tape scratching during the hydrogel treatment period.

Although not shown in this study, mechanical stimulation has been shown to be associated with inflammatory responses in a variety of immune cells associated with AD. For example, the receptors TRPV1, TRPV3, and TRPV4 on keratinocytes are involved in PAR2-mediated scratching and skin inflammation[10,14,60–62]. In particular, Asivatrep, a selective TRPV1 antagonist, has been shown to improve clinical symptoms in AD patients in phase III clinical trials[63]. However, the synergy, crosstalk, and interference between the numerous mechanosensors may lead to the poor therapeutic efficacy of single receptor blockers. As a convergence point for various mechanotransduction

pathways, FAK phosphorylation mediates inflammatory responses caused by mechanical stimuli. In this work, increased pFAK levels and inflammation were found in skin tissues of AD patients and AD mice, especially after mechanical scratching (Fig. 1). Our work re-emphasizes the importance of mechano-immunology for AD progression and suggests the potential of FAK as the target in the treatment of AD. Furthermore, given the recent findings regarding the role of mechanosensitive channel piezo in pruritic transmission[12], modulation of mechanoimmunity in pruritus could be a potential approach for AD treatment as well.

In addition to modulating mechano-immunology, FAK inhibition may treat AD by enhancing epidermal structural stability. Intercellular junctions are essential for maintaining the stability of epidermal structures and performing barrier functions. FAK-mediated focal adhesion has been widely shown to be competitive with E-cadherin-mediated intercellular adhesion[64,65], and inhibition of FAK can enhance the expression of E-cadherin[66], thereby enhancing intercellular junctions, which was verified in this work (Fig. 5f). Despite the lack of validation, another possible mechanism is that inhibition of FAK may enhance the mechanical compliance of cells, which reduces the physical disruption of intercellular junctions under scratching. FAK can affect cell stiffness by modulating cytoskeletal remodeling kinetic and contraction force[67], and the knockdown of FAK has been shown to increase cell invasion ability by decreasing cell stiffness and increasing cell compliance[68]. In AD treatment, enhanced cellular compliance may reduce excessive stress on intercellular junctions as caused by scratching. In this work, inhibition of cellular contractility by Blebbistatin rescued E-cadherin deficiency in cellular experiments, suggesting the therapeutic potential of cellular contraction inhibition for related diseases. However, existing studies on cellular contractility inhibitors, represented by myosin inhibitors, have focused on the cellular level, with few studies for animal treatment. Thus, the biosafety of related therapies needs further validation[69]. Besides, Blebbistatin has phototoxicity under visible light, which is detrimental to the treatment of skin[70]. In contrast, FAK inhibitors have been increasingly used in therapeutic studies of tumors and fibrotic scars[71,72]. Our results also highlight the clinical translational promise of FAK as a mechanotherapy target.

Controlled mechanical scratching may be an important but easily neglected variable in AD animal models. In most AD models, animal skins do not receive additional protection or mechanical intervention after exposure to the stimulus source[73]. Although animals will spontaneously scratch the damaged area because of itching, the scratching area and scratching frequency are uncontrollable. Besides, due to the positive feedback mechanism of the itch-scratch cycle for AD development, uncontrolled scratching may lead to regional differences in skin lesions. On the other hand, reduced scratching frequency due to the action of psychotropic drugs may reduce the scratching frequency of animals, further amplifying the therapeutic effect of the drug and leading to an inaccurate assessment of efficacy. In our AD model, tape tearing was used to simulate steady mechanical stimulation while the lesioned skin was covered with gauze to avoid spontaneous scratching by the mice. Our results indicate that tape tearing induces more severe skin lesions and inflammation than MC 903 application alone, highlighting the importance of mechanical stimulation in AD modeling (Fig. 2).

In summary, we have identified that FAK phosphorylation acts as a potential mechano-target for the treatment of AD and developed a skin dressing based on HCPF hydrogel to treat AD through mechano-chemically synergistic intervention. The approach of modulating oxidative stress by scavenging ROS and blocking mechanotransduction by inhibiting FAK targets the two major inducements in AD progression. The results indicate that HCPF hydrogels could achieve optimal inflammation inhibition and skin barrier repair, serving as a promising dressing for AD management (Fig. 7). Future works require deeper insights into the mechanisms of the coupling effects of mechanical stimulation and autoimmune on AD development, and thus inspiring therapeutic targets and solutions.

## Methods

### Materials

Carboxymethyl chitosan (CMCS), dopamine hydrochloride, ethyl-dimethyl-aminopropyl carbodiimide (EDC), N−hydroxy-succinimide (NHS), 4-carboxyphenylboronic acid (PBA), Egg PC, cholesterol, lauric acid, $NH_3 \cdot H_2O$, ethanol, and Nile red were purchased from Aladdin (China). Hyaluronic acids were purchased from Meilunbio (China). Terephthalic acid, dopamine hydrochloride, riboflavin, nitro blue tetrazolium, and methionine were purchased from Energy Chemical (China). Defactinib (VS-6063) and Calcipotriol (MC 903) were purchased from MedChemExpress, USA. anti-FAK (#71433), anti-Phospho-FAK (Tyr397, #8556), anti-β-actin (#4970), anti-E-cadherin (#3195), Anti-rabbit IgG (H+L, Alexa Fluor® 488 Conjugate, #4412), and anti-rabbit IgG HRP-linked secondary antibody (#7074S) were purchased from Cell Signaling Technology (USA). Anti-Filaggrin (ab221155) was purchased from Abcam (UK).

### Synthesis of liposomes

LipoLA was prepared by a vesicle extrusion technique following published procedures. In brief, liposomes composed of 14 mg of Egg PC, cholesterol, and lauric acid (7:1:2, weight ratio, respectively) were dissolved in 4 ml of chloroform and evaporated under nitrogen gas. Then, the dried lipid film was stored overnight under high vacuum to completely remove the chloroform. The dried lipid films were then rehydrated with 4 ml of deionized water. The suspended lipid was vortexed and then sonicated in a bath sonicator to produce multi-lamellar vesicles (MLVs). A Ti-probe was further used to produce small unilamellar vesicles (SUVs). Finally, the resulting SUVs were extruded through a 100-nm pore-sized polycarbonate membrane for 11 times with a Mini Extruder (Avanti Polar Lipids, Alabaster, AL, USA) to obtain the homogenous lipoLA. For loading the drug into the lipoLA, the Defactinib was mixed with the preformed liposomes at a concentration of 1 mM. Unencapsulated drugs were then removed from the liposomes by low-speed centrifugation method (500 rpm, 5 min) and then homogenized by the vesicle extrusion technique to obtain purified drug-loaded liposomes.

### Synthesis of PDA NPs

PDA NPs were synthesized according to a classical Stöber method. In brief, a mixture of concentrated $NH_3 \cdot H_2O$ (4 ml), ethanol (80 ml), and water (180 ml) was stirred at room temperature for 0.5 h, then aqueous dopamine hydrochloride (1 g in 20 ml $H_2O$) was added into the mixture. After stirring for 1 day, PDA NPs were harvested by centrifugation, washed with water, and freeze-dried.

### Characterization of liposomes and PDA NPs

The hydrodynamic diameters and zeta potential of the lipoLA, FAKi-lipoLA, and PDA NPs were measured using the Malvern Zetasizer ZS (Malvern Instruments, UK). The particle size distribution of liposomes and PDA NPs were statistically analyzed using ImageJ software through a transmission electron microscope (TEM, FEI Talos F200C) images. The loading yield of FAKi in lipoLA formulation was determined by HPLC. The filtered FAKi-lipoLAs were freeze-dried with the vacuum freeze dryer and redissolved in an equal volume of methanol prior to injection into the HPLC. The samples (100 μl) were loaded on a Thermo Scientific™ UltiMate™ 3000 instrument. HPLC method: mobile phase A (water) and mobile phase B (MeCN), flow rate: 1 ml/min, running time: 25 min, the gradient elution method: 95% B from 0 to 15 min, 95% to 5% B from 15 to 20 min. Detection wavelength: 254 nm. Injection volume:

10 µl. The percent encapsulation efficiency (EE) was calculated as shown below:

$$EE(\%) = \frac{\text{Amount of drug in liposomes}}{\text{Amount of drug used to make liposomes}} * 100\% \quad (1)$$

### Synthesis of HA-DA and CMCS-PBA and hydrogels preparation

HA-DA and CMCS-PBA synthesized through amidation reaction. To synthesize HA-DA, HA (1 g) was dissolved in 100 ml PBS followed by 0.5 g dopamine hydrochloride, 0.5 g EDC, and 0.36 g NHS added in the solution. The solution was stirred for 24 h, and the pH was maintained from 4.5 to 5 using 1 M NaOH. After the reaction, the solution was purified by dialysis against PBS (pH = 4) for 2 days and DI water (pH = 4) for 3 days. The obtained solution was adjusted to pH 7 using NaOH and lyophilized. To synthesize CMCS-PBA, CMCS was dissolved in 90 ml $H_2O$/DFM (3:2, v/v). Then, 0.4 g PBA, 0.46 g EDC, and 0.33 g NHS were dissolved in 20 ml of $H_2O$/DFM in a centrifuge tube. The mixture was allowed to react for 30 min at RT to activate the carboxyl groups and then added to the CMCS solution. The solution pH was maintained from 5 to 5.5 for 24 h. After the reaction, the solution was purified by dialysis against 1% $Na_2CO_3$ solution for 2 days and DI water for 3 days, then lyophilized. The successful grafting was confirmed by proton nuclear magnetic resonance (1H NMR, 400 MHz JEOL JNM-ECZ400S/L1). Blank HA-CS hydrogels can be simply formed by mixing HA-DA (4% w/v) and CMCS-PBA (4% w/v) precursors in equal volumes. To construct hydrogels loaded with different particles, PDA NPs, lipoLA, and FAKi-lipoLA was introduced HA-DA precursors then mixed with CMCS-PBA.

### Mechanical characterization

Rheology tests were carried out using an MCR 302 rheometer (Anton Paar, Austria). Samples were prepared at a thickness of 1 mm and diameter of 15 mm. Stretch tests were carried out using a universal testing machine (Wance, China) at a stretching rate of 1 mm/s. Samples were prepared in pieces of $30 \times 10 \times 1$ mm.

### Cell culture

Human immortalized keratinocyte HaCaT cells were purchased from commercial sources (ICell Bioscience Inc, Shanghai), which were cultured in low glucose Dulbecco's Modified Eagle Medium (DMEM, Gibco, USA) supplemented with 10% certified fetal bovine serum (Biological Industries, Israel), 100 µg/ml of streptomycin and 100 U/ml of penicillin (Gibco, USA). Cells were incubated at 37 °C with 5% $CO_2$ in a humidified atmosphere. The biocompatibility of hydrogels was evaluated by Calcein-AM/PI staining (Beyotime Biotechnology, China) and methyl thiazolyl tetrazolium (MTT) assay (Beyotime Biotechnology, China). In vitro stretch experiments were carried out using a custom-made stretching device. HaCaT cells were first seeded on a poly-dimethylsiloxane (PDMS, DOWSIL, USA) film for 24 h before stretching. Then the culture medium was replaced with hydrogel extract (1 ml hydrogel soaking in 10 ml serum-free medium for 24 h). After culturing for 12 h, cells were then stretched at different strain levels. After stretching, the conditioned medium was obtained for ELISA tests, and cells were fixed using 4% paraformaldehyde for immunofluorescence staining. E-cadherin deficiency is calculated as shown below:

$$E-cadherin\ deficiency = \frac{Number\ of\ E-cadherin\ uncolored\ junctions}{Total\ number\ of\ intercellular\ junctions} \times 100\% \quad (2)$$

### Antioxidant efficiency

HO· scavenging ability of PDA NPs was evaluated by detecting fluorescent 2-hydroxyterephthalic acid formed by terephthalic acid oxidation. $O_2^{·-}$ scavenging ability of PDA NPs was evaluated by nitro blue tetrazolium (NBT) assay (Beyotime Biotechnology, China). The total antioxidant capacity of hydrogel extracts was evaluated by 2,2'-azino-bis(3-ethylbenzthiazoline-6-sulfonic acid (ABTS) assay (Beyotime Biotechnology, China). Briefly, 100 mg HA-CS/PDA hydrogels with different PDA NPs concentrations were immersed in 1 ml RPMI-1640 for 12 h. The hydrogel extracts were tested using a total antioxidant capacity assay kit (Beyotime Biotechnology, China) according to the manufacturer's instructions. In vivo intracellular ROS scavenging ability of hydrogels was evaluated by using a reactive oxygen species assay kit (Beyotime Biotechnology, China). HaCaT cells were seeded in a 6-well plate fed with serum-free medium or co-incubation with HCPF hydrogels. Rosup reagent (1 µg/ml, 30 min) was used to raise the intracellular ROS levels. The intracellular ROS were detected by fluorescent DCFH-DA label.

### In vitro antibacterial efficiency of hydrogels

*Staphylococcus aureus* (*S. aureus*) model was used to evaluate the antibacterial activity of different HA-CS hydrogels. Briefly, 200 µl hydrogels were immersed into 1 ml bacteria suspension ($1.0 \times 10^7$ CFU/ml) and incubated at 37 °C for 6 h. Afterward, the bacterial solution was diluted and seeded on Luria Bertani (LB, Solarbio, China) broth agar plates, or the absorbance of bacterial suspension ($OD_{600}$) was measured. After cultivation for 12 h, bacterial colonies were calculated by the colony count of the agar plates.

### Assessment of FAK-lipoLA release

Nile red-labeled FAK-lipoLA was used to prepare HCF hydrogels. HCF hydrogels were made into thick discs 10 mm in diameter and 1 mm in thickness and placed in a transwell with polycarbonate membrane (pore size: 8 µm, Corning, USA). Transwell was placed in a 12-well plate, and DMEM was added to the plate well so that the liquid level of the medium in the transwell was approximately 0.5 mm. The medium was aspirated at different time points, and the fluorescence intensity of the medium was tested using a microplate reader to calculate the release amount of FAK-lipoLA.

### Animal experiment

All procedures were approved by the Xi'an Jiaotong University Biomedical Ethics Committee (No. 2022-1034). C57BL/6J male mice of age 6 weeks and were commercially purchased from Xi'an Jiaotong University laboratory animal center. Mice were group-housed with the parent mouse on a 12-h light-dark cycle at 22 °C with 50% humidity in the air. Mice were shaved and treated with 45 µM MC 903 (dissolved in anhydrous ethanol) and mechanical scratching for 14 days (scheme in Fig. 6a). Briefly, after anesthetizing by 2% isoflurane, the back skins of mice were tape-stripped with adhesive tapes (TegadermTM, 3M) for 10 times every day and then 40 µl 45 µM MC 903 were applied dropwise to the skin via pipette. Since the drug is dissolved in anhydrous ethanol, the drug solution will be rapidly absorbed after application. During the modeling process, the injured area was constantly covered with gauze to avoid mechanical stimulation caused by the mice scratching. After 14 days, significant AD skin lesions were observed, and mice were randomly divided into five groups. For the hydrogel treatment groups, skin lesions were covered with different hydrogels and applied 40 µl 20 µM MC 903 and six times of tape tears daily. Hydrogel dressings were removed prior to tape tear and MC 903 treatment and replaced with new dressings after treatment. For the untreated group, only 40 µl 20 µM MC 903 and six times of tape tears were administered. In all groups, mice were covered with gauze as before. Then, 24 h after the last administration, the skin lesions on the back of mice in each group were observed according to the SCORAD scoring standard, including erythema/bleeding, dry, desquamation, lichenification, etc., and graded as follows: none (score 0), mild (score 1), moderate (score 2), severe (score 3), and extremely severe (score 4).

The grading doctor is blind to the mice grouping. After 10 days of treatment, mice were euthanized, and tissues were obtained. To verify the in vivo antimicrobial efficiency of the hydrogel, we colonized *S. aureus* on the skin surface by applying gauze containing $1 \times 10^6$ CFU of *S. aureus* 2 days before treatment[74]. To determine bacterial numbers in the colonized skin, the skin of mice was scraped 20 times with cotton swabs to collect the colonized bacteria. Swabs were soaked in PBS for 1 h, and bacterial suspensions were inoculated in serial dilutions on LB broth agar plates.

### Human skin samples
Human skin sections were obtained from The Second Affiliated Hospital of Xi'an Jiaotong University. Normal human skin tissues were obtained from the edge of the excised nevus (four males, average age: 47.5 and four females, average age: 50.5). AD skin tissues were collected from eight patients (four males, average age: 40.75 and four females, average age: 53.25). All participants informed and consented to the experiment with no compensation. The use of paraffin-embedded skin sections was approved by the Xi'an Jiaotong University Biomedical Ethics Committee (No. 2022-1034).

### Histological assessment
Paraffin-embedded sections and H&E staining were performed to observe the epidermal thickness and the number of mast cells in the skin lesions of each group. The total number of mast cells in five high-power fields (400 times) of each mouse was calculated.

### Immunofluorescence and immunohistochemistry
Slides were first hydrated, and antigen retrieval was performed using sodium citrate. For immunofluorescence staining, slides were incubated with different primary antibodies at 4 °C overnight. The excess primary antibody was washed three times with PBS and incubated with an appropriate secondary antibody for 1 h and 4′,6-diamidino-2-phenylindole (DAPI, Thermo, USA) for 5 min at room temperature. Finally, all slides were covered with an antifade mounting medium and coverslips to reduce fluorescence attenuation. For immunohistochemistry staining, slides were incubated with horseradish peroxidase–conjugated rabbit anti-mouse IgG at room temperature for 20 min after incubation of primary antibodies. After washing, slides were treated with diaminobenzidine for 5–10 min and washed. For relative IOD level analysis, the epidermal intensity of immunostained pFAK and filaggrin was rated by Image Pro Plus application (version 6.0). All fluorescence intensities and IOD levels were subtracted from the values of the IgG control.

### ELISA assay
The skin lesion tissue homogenate and eyeball blood supernatant from experimental mice were obtained first, then protein expression levels of CCL-20, TNF-α, TSLP, IgE, IL-4, and IL-13 secreted into the supernatant were quantified in triplicate using ELISA kits (Biovision, China) following the manufacturer's instructions.

### Western blotting
Cells were lysed in RIPA Lysis Buffer containing protease inhibitor cocktail (P1005, Beyotime) at 1:100 dilution and separated by centrifugation for 10 min at 12,000 rpm. The supernatants were stored at −80 °C. Protein concentrations were determined on a Synergy Mx monochromator-based multi-mode microplate reader (Biotek, USA) using a BCA Protein Kit (Thermo Scientific, USA). Total protein was separated by 10% SDS-PAGE and transferred to PVDF membranes (Millipore, USA). Blots were treated with 5% BSA (Millipore, USA) for 1 h and incubated with primary antibody in 5 ml blocking buffer at 4 °C overnight with rotation, followed by incubation with secondary antibody at room temperature with rotation for 1 h. Normalization was performed by blotting the same membranes with anti-β-actin. The bands were detected using Ultra High Sensitivity ECL Kit (MCE, 5 ml) and imaged on a ChemiScope 3000mini Imaging System (Clinx) using automated exposure settings. Data are the average of three biological replicates unless indicated otherwise.

### Statistics
Statistical analyses were performed using GraphPad Prism 9.0. Images are processed by ImageJ (NIH, USA). Confocal images were obtained with the Olympus FV31S-SW software on an Olympus FV3000 microscope. Unless specifically mentioned, a two-tailed Student's *t*-test was used to compare differences between the two groups. One-way analysis of variance (ANOVA) with Bonferroni post hoc testing was used to carry out multiple comparisons among the three or more groups. $P < 0.05$ was taken as the threshold for significant differences. Materials experiments were performed by replicating tests of multiple samples, and data are shown as mean ± s.d. Cell experiments were replicated over multiple experiments, and data are shown as mean ± s.e.m. For in vivo experiments, each slice used for quantification was obtained from different mice, and data are shown as mean ± s.e.m.

### Reporting summary
Further information on research design is available in the Nature Portfolio Reporting Summary linked to this article.

## Data availability
All relevant data supporting the key findings of this study are available within the article and its Supplementary Information files or from the corresponding author upon reasonable request. Source data are provided with this paper.

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

## Acknowledgements

This work was financially supported by the National Natural Science Foundation of China (12225208, 11972280, 82273526) and the Shaanxi Province Innovation Team (2022TD-48).

## Author contributions

Y.J., S.G., and F.X. conceived the ideas and designed the experiments. J.H., F.X., and S.G. directed the animal experiments. Y.J., J.H., K.A., Q.Z., Y.D., H.L., and Z.W. conducted the experiments and analyzed the data. All authors interpreted data and contributed to the writing of the manuscript.

## Competing interests

The authors declare no competing interests.
