## [Peer Review File · Nature Communications]

REVIEWER COMMENTS

Reviewer #1 (Remarks to the Author):

Jia et al. present a hydrogel dressing for atopic dermatitis (AD). The hydrogel is loaded with (i) polydopamine nanoparticles (PDA NPs) to scavenge reactive oxygen species, and (ii) liposome-encapsulated FAK inhibitors (FAKi-lipo) to reduce epithelial junction deficiency mainly caused by mechanical loading such as scratching. The topic of the study is of great importance and the results show that the proposed approach can be effective to alleviate AD symptoms. The methodology is sound and the work will be of significance to the field. Below, please find my comments which I hope can further improve the quality of the work.

1. "FAK is at the point of biological convergence of various signaling pathways"

Although FAK plays an essential role in the mechanobiological behavior of cells, however, I am not sure if it is "the point of biological convergence of various signaling pathways". Although the results here show that inhibitions of FAK can reduce the mechanobiological response of cells to mechanical deformations, I expect to see a similar behavior upon inhibition of actomyosin contractility. Indeed, several studies have shown that inhibition of FAK reduces actomyosin contractility. Therefore, adding any inhibitor of actomyosin contractility is expected to decrease the tension at the cell-cell junctions and subsequently reduce the possibility of rupture at cell-cell junctions.

2. How many normal and AD human samples were used in figure 2b? Please provide more H&E images of normal and AD human samples containing the epidermis and part of the dermis as shown in figure 2a.

3. For the in vitro studies in Figure 5, did you place HCF hydrogel on HaCaT cells? Could you please provide an image of that?

4. If you placed an HCF hydrogel on HaCaT cells for the stretching experiments in Figure 5C, did you also place a blank hydrogel (HC) on cells for the control case?

5. "We observed that CCL-20 and TSLP secretions are significantly increased in the mechanical stretch group relative to the unstretched group (control), while treating with HCF hydrogel significantly reduces their expressions (Fig. 5d)."

I do not see CCL-20 measurements in Figure 4d. Did you measure the protein expression levels of both CCL-20, and TNF?

6. I think it needs to be mentioned that although treatment with the hydrogel dressing alleviates AD symptoms, however, epidermal thickness (figure S6), TSLP level (figure 7b), and IL-4 level (figure 7d) are still significantly higher than in healthy non-AD cases.

7. Do you have a comparison between control and AD with no scratching for IgE and IL-4, as well as the protein levels of TSLP, CCL-20, and IL-13?

8. It has been already shown that scavenging reactive oxygen species can alleviate AD symptoms. Therefore, I expected to see more comparisons between AD with and without mechanical deformations to show that FAK inhibition can further reduce the symptoms. However, in some cases, there is no comparison between AD with and without mechanical deformations. For example, why in Figure 6, there is no AD without scratch?

9. In figure 7f, it seems that there is no decrease in E-cadherin fluorescence intensity compared with AD_scratch. Could you please clarify?

10. "liposome encapsulation has been used to promote the drug loading and sustained release of hydrophobic FAKi"
Please provide references.

11. Please, mention that the "frequency sweep test" refers to shear testing using a rheometer.

Farid Alisafaei, Ph.D.
New Jersey Institute of Technology (NJIT)

Reviewer #2 (Remarks to the Author):

In this manuscript, Jia and colleagues developed an antibacterial self-healing hydrogel dressing that releases FAK inhibitors and ROS scavengers to treat atopic dermatitis. While ROS is known to be implicated in AD, this work shows for the first time that FAK signaling in the epidermis is activated by skin scratching, which led the authors to hypothesize that combining FAK inhibitors to dampen aberrant FAK signaling caused by scratching with ROS scavengers to reduce the inflammatory responses may be more effective strategy to treat AD compared to single factor treatments. Overall, the experiments are well designed and executed, and the in vivo data is compelling that this multifactorial approach is promising for treating this skin disorder. The manuscript is well written, but I have a few concerns and comments that need to be addressed to strengthen the findings and the impact of this study.

Major comments:

The methods on the animal experiments should be clarified. How was the MC-903 applied? Topically with a dressing, or by subcutaneous injection? After 14 days of injury stimulus, the skin lesions were covered with different hydrogels and 20 μ M MC-903 and 6 times of tape tears were applied daily. It's not clear if the hydrogels were removed from the lesion prior to the tape tears and MC-903 treatment, or if the gels remained in place. Also, was the same hydrogel used for the entire period of treatment, or was the hydrogel replaced with a fresh one daily?

The authors present in vitro data on the ROS scavenging properties of the hydrogels. But, given that the top keratinocyte layer is usually a dead layer of skin, how deep do the PDA nano particles penetrate the skin? Does scavenging ROS only manifest in the epidermis, or does it go beyond the basal epidermal layer?

It is not clear from the materials and methods how the release profile of FAKi was measured? Were the gels placed in an aqueous environment? Or were they placed in a transwell at a liquid interface, which would better approximate how the hydrogel releases its cargo when placed topically on the skin?

The antibacterial properties of the hydrogel are convincingly demonstrated in vitro, but to what extent does the hydrogel eradicate bacterial growth in the skin, for instance after inoculation of the skin lesion with *S. aureus*? Is the treatment FAKi/ROS scavengers effective when a biofilm is present?

In Figure 3 (e and f): The HaCAT monolayer looks immature given that only half of the field of view is covered with cells. Cadherin stability and expression is strongly affected by cell density. How does stretch and stretch + HCF affect E-cadherin expression/localization in confluent monolayers? Images of static controls are needed to better gauge the effect of stretch and hydrogel treatment. How is E-cad deficiency measured?

Although the authors mention n values for each experiment, it is unclear whether 'n' stands for the number of technical replicates (i.e. duplicates or triplicates in one experiment) or biological replicates

(i.e. different experiments). Given that standard deviation (SD) is reported rather than standard error of the mean (SEM), I presume that the in vitro experiments were performed once with n samples. The general consensus is to report data from at least three experimental repeats.

For in vivo experiments, each mouse can be considered as a separate experiment, which is why SEM rather than SD should be reported. Similarly, quantification of histological images should mention the number of sections/ROIs from how many mice that were used for the analysis.

The histology in figure 1, 6 and 7 is well executed, but the 'high magnification' images are too low in magnification to provide any additional informative compared to the low magnification images. The high magnification images should at least be two-fold higher or more in magnification as to better show where the signal for pFAK and fillagrin is at the cellular level. Also, arrows indicating positive signal will be helpful to the readers who aren't familiar with DAB stainings. Lastly, IGG controls for IF and IHC staining should be included, and the methods of quantification of the staining should be clarified.

Minor Comment

Can the authors elaborate on the self-healing properties of the hydrogel and include a few sentences on the mechanism of 'self-healing' works?

Figure 7: A red line is used to demarcate the deficient area from the intact area, which is difficult to see for any reader whose color vision is impaired. A white line would be a better choice.

Line 76: What is mechanosensitization? I'm not sure if this word exists.

Typo's that need to be corrected:

Line 17: synergistally

Line 136 - "Lauri Acid" mistyped?

Fig 6a - "Tap Scractch" mistyped?

Throughout the paper, 'tap tears' should be 'tape tears'

Reviewer #3 (Remarks to the Author):

The manuscript by Jia et al, details the development of a hydrogel dressing for the potential use with atopic dermatitis.

Before the manuscript can be considered for publication there are a number of items that should be addressed.

Line 45: 'Mechanical scratch is an important cause leading to ...' how this sentence is worded does not make sense, please reword.

Line 84: the statement regarding hydrogels and being used for their good biocompatibility. Using the term biocompatibility in general like this is incorrect. Biocompatible refers to the use of a certain material within a specific application. The general term of hydrogel does not specify the material used, not all hydrogels are biocompatible when used as in a skin dressing application.

Line 110: refers to patients, this gives the impression that experiments were carried out in humans rather than mice. Please amend.

Line 115: refers to rat scratching, though in vivo experiments were carried out in mice

Line 119: refers to mast cell infiltration, through these sections were not stained to evaluate mast cell evaluation, please modify statement

Line 174: states that the results show the hydrogels can be removed painlessly, though results shown do not demonstrate level of pain felt when hydrogels are removed from skin

Line 272: refers to the expression of CCL-20 and TSLP (Fig 5d) though Fig 5d shows the expression of TNF- α and TSLP, please amend text.

The text that is referring to Fig 6 is incorrect, example, line 305/306 refers to mast cell infiltration by toluidine blue staining (Fig 6e,f) though Fig 6f shows epidermal thickness.

Line 337: text states (Fig h,i) please include figure number.

Response Letter

Re: # NCOMMS-22-45011A

Title: A Multifunctional Hydrogel Dressing Integrating FAK Inhibition and ROS Scavenging for Mechano-chemically Synergistic Treatment of Atopic Dermatitis

Dear Reviewers,

We would like to thank you for your time to review our manuscript and your invaluable and constructive comments. We have revised the manuscript according to the comments and significantly improved the quality of our paper. For convenience, the reviewers' comments have been marked in our current response letter *in nattier blue and italic font* and the response to the comment is in black. Changes in the manuscript are marked in **red**.

Response to Reviewer #1

Jia et al. present a hydrogel dressing for atopic dermatitis (AD). The hydrogel is loaded with (i) polydopamine nanoparticles (PDA NPs) to scavenge reactive oxygen species, and (ii) liposome-encapsulated FAK inhibitors (FAKilipo) to reduce epithelial junction deficiency mainly caused by mechanical loading such as scratching. The topic of the study is of great importance and the results show that the proposed approach can be effective to alleviate AD symptoms. The methodology is sound and the work will be of significance to the field. Below, please find my comments which I hope can further improve the quality of the work.

Response: We thank the reviewer for the encouraging comment.

***Comment #1.** “FAK is at the point of biological convergence of various signaling pathways.” Although FAK plays an essential role in the mechanobiological behavior of cells, however, I am not sure if it is “the point of biological convergence of various signaling pathways”. Although the results here show that inhibitions of FAK can reduce the mechanobiological response of cells to mechanical deformations, I expect to see a similar behavior upon inhibition of actomyosin contractility. Indeed, several studies have shown that inhibition of FAK reduces actomyosin contractility. Therefore, adding any inhibitor of actomyosin contractility is expected to decrease the tension at the cell-cell junctions and subsequently reduce the possibility of rupture at cell-cell junctions.*

Response:

(1) We thank the reviewer for pointing out this. Many papers have highlighted FAK as a downstream convergence of mechanotransduction and many inflammatory signaling pathways. We have changed the vague description and added relevant references in the manuscript on **Page 6**.

On **Page 6**:

“FAK is at the point of biological convergence of mechanotransduction and many inflammatory signaling pathways¹⁶⁻¹⁸.”

(2) We agree with the reviewers. As suggested by the reviewers, we used the selective non-muscle myosin II (NMII) inhibitor Blebbistatin to inhibit actomyosin contractility of HaCaT and repeated the stretching experiments. Many papers have reported that excessive inhibition of actomyosin contractility leads to reduced cell spreading, which would directly impair intercellular junctions. Thus, we first screened the safe doses of FAK-lipoLA and Blebbistatin under our experimental conditions. The results showed that 1,000 $\mu\text{g/ml}$ of FAK-lipoLA (the maximum loading concentration of FAK-lipoLA in hydrogels) did not affect the morphology of HaCaT cells. Treatment with 100 μM of Blebbistatin resulted in a significant reduction in cell spreading, and treatment with 50 μM of Blebbistatin also resulted in abnormal morphology of HaCaT cells. Therefore, in our experiments, 10 μM and 30 μM were chosen as working concentrations. Similar to the effect of FAK inhibition, inhibition of actomyosin contractility by Blebbistatin rescued the stretch-induced E-cadherin deficiency. However, compared to 30 μM , 10 μM Blebbistatin had a significantly reduced ability to rescue E-cadherin deficiency. These results suggest that inhibition of FAK may reduce the rupture of intercellular junctions under stretch conditions by reducing cellular contractility. It's also worth noting that existing studies on cellular contractility inhibitors, as represented by myosin inhibitors, mainly focused on the cellular level, with few studies for animal treatment. Thus, the biosafety of related therapies needs further validation. Besides, Blebbistatin has significant phototoxicity under visible light, which is detrimental to the treatment of skin. We have added the relevant results in **Supplementary Figure S6** and added related descriptions in the **Results** section on **Page 10** and in the **Discussion** section on **Page 15**.

On **Page 10**:

“Several studies have demonstrated that FAK inhibition decreases actin contractility⁵⁰. To verify whether FAK inhibition rescues E-cadherin deficiency by decreasing cellular contractility, we treated cells with the non-muscle myosin II inhibitor Blebbistatin. Blebbistatin above 50 μM will lead to abnormal cell morphology and even reduced cell spreading, which will directly impair intercellular junctions. In contrast, 1,000 $\mu\text{g/ml}$ of FAK-lipoLA (the maximum loading concentration of FAK-lipoLA in hydrogels) did not affect the morphology of HaCaT cells (**Supplementary Fig. S6a,b**). Therefore, 10

μM and $30 \mu\text{M}$ were chosen as working concentrations. Inhibition of actomyosin contractility by Blebbistatin rescues stretch-induced E-cadherin deficiency (Supplementary Fig. S6c,d). However, compared to $30 \mu\text{M}$, $10 \mu\text{M}$ Blebbistatin has a significantly reduced ability to rescue E-cadherin deficiency.”

on Page 15:

“In this work, inhibition of cellular contractility by Blebbistatin rescued E-cadherin deficiency in cellular experiments, suggesting the therapeutic potential of cellular contraction inhibition for related diseases. However, existing studies on cellular contractility inhibitors, represented by myosin inhibitors, mainly focused on the cellular level, with few studies for animal treatment⁶⁴. Thus, the biosafety of related therapies needs further validation. Besides, Blebbistatin has phototoxicity under visible light, which is detrimental to the treatment of skin⁶⁵. In contrast, FAK inhibitors have been widely used in therapeutic studies of tumors and fibrotic scars^{66,67}. Our results also highlight the clinical translational potential of FAK as a mechanotherapy target.”

Supplementary Figure S6 | Effect of blebbistatin on cell morphology and E-cadherin under stretch.

a-b, HaCaT cell morphology after treatment with different concentration of FAK-lipoLA and blebbistatin. Cells are stained with F-actin (red) and DAPI (blue). Scale bar, $100 \mu\text{m}$. **c**, Fluorescent staining of HaCaT cells after stretching with E-cadherin (green), F-actin (red) and DAPI (blue). Scale bar, $50 \mu\text{m}$. **d**, E-cadherin deficiency of HaCaT

cells calculated from staining images in **c**. $n = 4$. Data are shown as mean \pm s.e.m. and compared by one-way ANOVA followed by Bonferroni's post hoc test. *** and **** indicate $P < 0.001$ and $P < 0.0001$, respectively.

Comment #2. How many normal and AD human samples were used in figure 2b? Please provide more H&E images of normal and AD human samples containing the epidermis and part of the dermis as shown in figure 2a.

Response: We thank the reviewer for pointing out this. We used 8 tissue samples from different patients in each group. Following the suggestion, we have added a description of the number of samples in **Figure 2b** and presented the H&E staining images for all samples in **Supplementary Figure S1**.

Figure 2 | pFAK is upregulated in AD skin, especially after scratching.

b, Epidermal thickness quantified from H&E staining in **a**, For human skin, $n = 8$. For mouse skin, $n = 5$.

Supplementary Figure S1 | H&E staining of human skins.

a, H&E staining of skin sections. The black boxed area is enlarged below. The space between red lines denotes the epidermal thickness. Scale bar, 100 μm .

Comment #3. For the in vitro studies in Figure 5, did you place HCF hydrogel on HaCaT cells? Could you please provide an image of that?

Response: We are sorry for the confusion. We used hydrogel extract to culture the cells. As suggested by the reviewer, we have used blank hydrogel (HC) extract as a control group and repeated the relevant experiments. We have changed the schematic of **Figure 5c** and added relevant descriptions in the main text on **Page 10** and in **Materials and method** on **Page 19**.

On Page 10:

“To simulate the mechanical scratching *in vitro*, we used a customized stretching device to apply different cyclic stretching to HaCaT cells. Cells of the HC group (blank control, cultured with HC hydrogel extract) and the HCF group (cultured with HCF hydrogel extract) were separately seeded in the two compartments of a PDMS membrane to ensure that the two groups of cells are subjected to the same stretching (**Fig. 5c**).”

On Page 19:

“*In vitro* stretch experiments were carried out using a custom-made stretching device. HaCaT cells were first seeded on a polydimethylsiloxane (PDMS) film for 24 h before stretching. Then, the culture medium was replaced with hydrogel extract (1 ml hydrogel soaking in 10 ml serum-free medium for 24 h). After culture for 12 h, cells were stretched at different strain levels. After stretching, the conditioned medium was obtained for ELISA tests, and cells were fixed using 4% paraformaldehyde for immunofluorescence staining.”

Comment #4. If you placed an HCF hydrogel on HaCaT cells for the stretching experiments in Figure 5C, did you also place a blank hydrogel (HC) on cells for the control case?

Response: We are sorry for the confusion. We used HC hydrogel extract to culture HaCaT cells as the control group. To clarify this, we have revised related text.

Comment #5. “We observed that CCL-20 and TSLP secretions are significantly increased in the mechanical stretch group relative to the unstretched group (control), while treating with HCF hydrogel significantly reduces their expressions (Fig. 5d).” I do not see CCL-20 measurements in Figure 4d. Do you measure the protein expression levels of both CCL-20, and TNF?

Response: We thank the reviewer for pointing out this. We have added the experimental data of CCL-20 in **Figure 5d** and made corresponding changes in the main text on **Page 11**.

On **Page 11**:

“For HC group, we observed that TNF- α , TSLP and CCL-20 secretions are significantly increased after 30% stretching, while treatment with HCF extract significantly reduces their expressions under 30% stretching. Besides, for the 5% stretching groups, there is no difference between the HC and HCF groups, indicating that FAK-lipoLA does not affect the expression of inflammatory factors in the unstretched state (**Fig. 5e**).”

Figure 5 | Inhibition of FAK protects cells from mechanical damage *in vitro*.

e, TNF- α , TSLP and CCL-20 level in the culture medium of HaCaT cells after stretch.

Comment #6. I think it needs to be mentioned that although treatment with the hydrogel dressing alleviates AD symptoms, however, epidermal thickness (figure S6), TSLP level (figure 7b), and IL-4 level (figure 7d) are still significantly higher than in healthy non-AD cases.

Response: We agree to this. Following the reviewer’s suggestion, we have added a description in **Discussion** section on **Page 14**.

On **Page 14**:

“It should be mentioned that although treatment with hydrogels relieved AD symptoms, in the HCPF group, skin lesions (**Fig. 6**), inflammatory markers (such as TSLP and IL-4, **Fig. 7b,d**) and intercellular junctions (**Fig. 7g,i**) are still not restored to the level of healthy mice. One possible reason is that the mice were exposed to a little dose of MC 903 and tape scratching during the hydrogel treatment period.”

Comment #7. Do you have a comparison between control and AD with no scratching for IgE and IL-4, as well as the protein levels of TSLP, CCL-20, and IL-13?

Response: Following the reviewer’s suggestion, we have added a comparison between normal mice and AD with no scratching mice in **Supplementary Fig. S2c-g**.

Supplementary Figure S2 | Evaluation of inflammation and epidermal barrier damage after scratching AD skins.

c-g, CCL-20, TSLP, IgE, IL-4 and IL-13 levels in mouse tissues of each group, n= 5 mice. All data are shown as mean ± s.e.m.. *, **, ***and **** indicate $P < 0.05$, $P < 0.01$, $P < 0.001$ and $P < 0.0001$ compared using a two-tailed Student’s t-test, respectively.

Comment #8. It has been already shown that scavenging reactive oxygen species can alleviate AD symptoms. Therefore, I expected to see more comparisons between AD with and without mechanical deformations to show that FAK inhibition can further reduce the symptoms. However, in some cases, there is no comparison between AD with

and without mechanical deformations. For example, why in Figure 6, there is no AD without scratch?

Response: We thank the reviewer for pointing out this. Comparison of H&E, pFAK and inflammatory factor between AD with and without mechanical scratch has been shown in **Figure 1** and **Supplementary Fig. S2**. This narrative order is intended to make our story sound more logical, *i.e.*, we first discovered the important role of scratching on AD inflammation and epidermal barriers and then carried out detailed studies afterwards. On the other hand, all treatments in **Figure 6** were performed on AD mice with scratching. We think that it is not proper to compare these treatment results with scratching-free AD mice.

As suggested by the reviewers, we have added additional assessments of mast cell and epidermal barriers in **Supplementary Fig. S2**. There is no significant difference in mast cell infiltration in the AD_{scratch} group compared to the unscratched group. However, by comparing the E-cadherin staining, we observed that the epidermis of unscratched AD skin is thickened with high fluorescence intensity of E-cadherin throughout the epidermis, while the fluorescence intensity of E-cadherin is significantly stratified in scratched AD skin. The filaggrin expression in the stratum corneum is also higher in the AD group than in the AD_{scratch} group. These results indicate that scratching may cause the damage of epidermis barriers. To clarify this, we have added descriptions on **Page 5**.

On Page 5:

“Disruption of the epidermal barrier is one of the hallmarks of AD lesions and will increase skin sensitization to allergens. Thus, we evaluated two important skin barrier proteins, *i.e.*, E-cadherin (mainly expressed in the epidermis) and filaggrin (mainly expressed in the stratum corneum). Immunofluorescent staining of E-cadherin shows obvious epidermal thickening in the AD and AD_{scratch} group. Unscratched AD skin shows high fluorescence intensity of E-cadherin throughout the epidermis, while the fluorescence intensity of E-cadherin is significantly stratified in the AD_{scratch} group (**Supplementary Fig. S2h,i**). The filaggrin expression in the stratum corneum is also higher in the AD group than in the AD_{scratch} group (**Supplementary Fig. S2j-l**). Besides,

compared to healthy skin, pFAK is enriched in the AD group and significantly elevated in the epidermis of the AD_{scratch} group (Fig. 2d). These results indicate that scratching may cause the damage of epidermis barriers, and pFAK is associated with AD and significantly enhanced by mechanical scratching.”

Supplementary Figure S2 | Evaluation of inflammation and epidermal barrier damage after scratching AD skins.

a, Representative toluidine blue staining of skin section. The red triangle denotes dermal mast cells. Scale bar, 100 μm . $n = 5$ mice. **b**, Measurement of the density of mast cells for each group after treatments. **c-g**, CCL-20, TSLP, IgE, IL-4 and IL-13 levels in mouse tissues of each group, $n = 5$ mice. **h**, Immunofluorescent (IF) staining of E-cadherin (green) and DAPI (blue) in each group. Scale bar, 20 μm . The white lines denote the deficient area of E-cadherin in the epidermis. In the magnified images, the yellow arrows denote intact intercellular junctions represented by intact E-cadherin, while the white arrows indicate deficient E-cadherin. **i**, Quantification of tissue thickness with E-cadherin deficiency, $n = 4$ mice. **j**, Immunohistochemistry (IHC) staining of filaggrin in each group. Scale bar, 100 μm for the images above and 25 μm for the magnified images. The red arrows indicate the area of high DBA staining (high filaggrin expression). **k**, Relative IOD of filaggrin quantified from images of **h**, $n = 4$ mice. **l**, Representative images of IHC and IF staining of IgG. Scale bar, 50 μm . All data are shown as mean \pm s.e.m.. *, **, *** and **** indicate $P < 0.05$, $P < 0.01$, $P < 0.001$ and $P < 0.0001$ compared by one-way ANOVA followed by Bonferroni's post hoc test, respectively.

Comment #9. In figure 7f, it seems that there is no decrease in E-cadherin fluorescence intensity compared with AD_scratch. Could you please clarify?

Response: We apologize for the vague descriptions of the results. For a clear view of the E-cadherin deficiency, we have magnified the images and indicated the intact E-cadherin with yellow arrows and the defective E-cadherin with white arrows. As the reviewer mentioned, there is no difference in the fluorescence intensity of E-cadherin in the regions close to the dermis for all groups. However, the fluorescence intensity of E-cadherin is significantly lower in the regions close to the stratum corneum, while these regions are colored with nuclei. Therefore, we consider these regions as E-cadherin deficient regions. Furthermore, epidermal thickness as quantified by E-cadherin staining showed similar results to H&E staining (**Supplementary Fig. S8h**). This suggests that the region where the nucleus is colored but E-cadherin fluorescence intensity is low is part of the epidermis. To better present the E-cadherin-deficient part, we used white lines to outline the cuticle and the demarcation between the deficient area and the intact area of E-cadherin (**Fig. 7f**).

Figure 7 | HCPF hydrogels reduce inflammation and epidermal barrier damage of AD mice.

f, Fluorescent staining of E-cadherin (green) and DAPI (blue) in each group. Scale bar, 20 μm . The white lines denote the deficient area of E-cadherin in the epidermis. In the magnified images, the yellow arrows denote intact intercellular junctions represented by intact E-cadherin, while the white arrows indicate defective E-cadherin.

Comment #10. “liposome encapsulation has been used to promote the drug loading and sustained release of hydrophobic FAKi” Please provide references.

Response: We thank the reviewer for pointing out this wrong description. There has been no report on liposome encapsulated FAK inhibitor. Our intent was that liposome encapsulation has been used to promote the drug loading and sustained release of hydrophobic drugs. We have changed the description and added references on **Page 4**.

On **Page 4**:

“while liposome encapsulation has been used to promote the drug loading and sustained release of hydrophobic drugs^{33,34}”

Comment #11. Please, mention that the “frequency sweep test” refers to shear testing using a rheometer.

Response: We thank the reviewer for pointing out this. We have added this in the manuscript on **Page 7**.

On **Page 7**:

“Frequency sweep of shearing test by a rheometer revealed that the HC hydrogels have remarkable viscoelastic properties with a storage modulus of 1.24 ± 0.17 kPa, which is smaller than the modulus of human skin⁴¹.”

Response to Reviewer #2

In this manuscript, Jia and colleagues developed an antibacterial self-healing hydrogel dressing that releases FAK inhibitors and ROS scavengers to treat atopic dermatitis. While ROS is known to be implicated in AD, this work shows for the first time that FAK signaling in the epidermis is activated by skin scratching, which led the authors to hypothesize that combining FAK inhibitors to dampen aberrant FAK signaling caused by scratching with ROS scavengers to reduce the inflammatory responses may be more effective strategy to treat AD compared to single factor treatments. Overall, the experiments are well designed and executed, and the in vivo data is compelling that this multifactorial approach is promising for treating this skin disorder. The manuscript is well written, but I have a few concerns and comments that need to be addressed to strengthen the findings and the impact of this study.

Response: We thank the reviewer for the encouraging comment.

Major comments:

***Comment #1.** The methods on the animal experiments should be clarified. How was the MC-903 applied? Topically with a dressing, or by subcutaneous injection? After 14 days of injury stimulus, the skin lesions were covered with different hydrogels and 20 μ M MC-903 and 6 times of tape tears were applied daily. It's not clear if the hydrogels were removed from the lesion prior to the tape tears and MC-903 treatment, or if the gels remained in place. Also, was the same hydrogel used for the entire period of treatment, or was the hydrogel replaced with a fresh one daily?*

Response: We thank the reviewer for pointing out this. The MC 903 was directly applied dropwise to skin via pipette. Since the drug is dissolved by anhydrous ethanol, the drug solution will be rapidly absorbed after application. The hydrogels were removed from the lesion prior to the tape tears and MC 903 treatment and replaced with a fresh one daily. We have clarified these in the methods on **Page 20**.

On **Page 20**:

“6- to 8-week-old male mice was shaved and treated with 45 μ M MC 903 (dissolved in

anhydrous ethanol) and mechanical scratching for 14 days (scheme in Fig. 6a). Briefly, after anesthetizing by 2% isoflurane, back skins of mice were tape-stripped with adhesive tapes (Tegaderm™, 3M) for 10 times every day and then 40 µl 45 µM MC 903 were applied dropwise to skin via pipette. Since the drug is dissolved by anhydrous ethanol, the drug solution will be rapidly absorbed after application. During the modeling process, the injured area was constantly covered with gauze to avoid mechanical stimulation caused by the mice scratching. After 14 days, significant AD skin lesions were observed, and mice were randomly divided into 5 groups. For the hydrogel treatment groups, skin lesions were covered with different hydrogels and applied 40 µl 20 µM MC 903 and 6 times of tape tears daily. Hydrogel dressings were removed prior to tape tear and MC 903 treatment and replaced with new dressings after treatment.”

Comment #2. The authors present in vitro data on the ROS scavenging properties of the hydrogels. But, given that the top keratinocyte layer is usually a dead layer of skin, how deep do the PDA nano particles penetrate the skin? Does scavenging ROS only manifest in the epidermis, or does it go beyond the basal epidermal layer?

Response: We thank the reviewer for pointing out this. To test the *in vivo* ROS scavenging effect of hydrogels, we stained whole skin sections for 8-OHdG. 8-OHdG is an oxidation product of deoxyguanosine residues in DNA, formed by the attack of the hydroxyl group ($\cdot\text{OH}$) at the C-8 position of guanine, and has been widely used as a marker of oxidative cellular damage. Relative to the untreated AD_{scratch} group, the fluorescence intensity of 8-OHdG is decreased in both the epidermis and dermis in the PDA-loaded HCP and HCPF groups, indicating that the hydrogels can effectively scavenge ROS in both dermis and epidermis. To clarify these, we have added relevant results in **Supplementary Figure S8**, and the following text on **Page 12** of the main text.

Supplementary Fig. S8 | Evaluation of serum factors and oxidative damage in tissues.

f, Fluorescent staining of 8-OHdG (green) and DAPI (blue) in each group. Scale bar = 50 μ m. **g**, Fluorescence intensity of 8-OHdG reveals oxidative DNA damage in each group.

On Page 12:

“Besides, to test the *in vivo* ROS scavenging effect of hydrogels, fluorescent staining was performed on slides to detect 8-hydroxy-2'-deoxyguanosine (8-OHdG) (Supplementary Fig. S8f), an oxidation product of deoxyguanosine residues in DNA^{1,2}. The results showed that HCP and HCPF hydrogels loaded with PDA NPs could effectively reduce DNA damage in the epidermis and dermis (Supplementary Fig. S8g). The fluorescence intensity is slightly reduced in the HC and HCF groups relative to the AD_{scratch} group, which may be due to the ROS scavenging effect of the dopamine groups in the HC hydrogels and the reduced acute inflammation by FAK inhibition.”

Comment #3. It is not clear from the materials and methods how the release profile of FAKi was measured? Were the gels placed in an aqueous environment? Or were they placed in a transwell at a liquid interface, which would better approximate how the hydrogel releases its cargo when place topically on the skin?

Response: We thank the reviewer for pointing out this. We tested the release of FAK-lipoLA by placing the HCF hydrogel in a transwell at a liquid interface as suggested by the reviewer. To clarify this, we have revised relevant results in **Figure 5b** and added relevant descriptions in main text on **Page 10** and **Page 20**.

On Page 10:

“We placed HCF hydrogels in a transwell to simulate the skin-hydrogel interface. The result showed that the FAKi-lipoLA in HCF could be released about 13% at 24 h and 18% at 72 h (Fig. 5b).”

On Page 19:

“**Assessment of FAK-lipoLA release.** Nile red-labeled FAK-lipoLA was used to prepare HCF hydrogels. HCF hydrogels were made into thick discs with 10 mm in diameter and 1 mm in thickness and placed in a transwell with polycarbonate membrane (pore size: 8 μm , Corning). Transwell was placed in a 12-well plate and DMEM was added to the plate well so that the liquid level of the medium in the transwell was approximately 0.5 mm. The medium was aspirated at different time points, and the fluorescence intensity of medium was tested using a microplate reader to calculate the release amount of FAK-lipoLA.”

Figure 5 | Inhibition of FAK protects cells from mechanical damage *in vitro*.

b, Time-related FAK-lipoLA release from HCF hydrogels placed in transwell. Data are shown as mean \pm s.d..

Comment #4. The antibacterial properties of the hydrogel are convincingly demonstrated in vitro, but to what extent does the hydrogel eradicate bacterial growth in the skin, for instance after inoculation of the skin lesion with S. aureus? Is the treatment FAKi/ROS scavengers effective when a biofilm is present?

Response: We thank the reviewer for pointing out this. To verify the *in vivo* antimicrobial efficiency of the hydrogel, we have incubated *S. aureus* on the skin surface of AD mice by applying gauze containing *S. aureus* two days before treatment.

The number of bacteria on the skin was assessed on day 0 and 10 of treatment, respectively. The results showed that HCPF hydrogel could effectively inhibit *S. aureus* in the skin of mice and effectively alleviate AD symptoms under bacterial infection conditions. We have added relevant results in **Supplementary Figure S8** and relevant descriptions in main text on **Page 13** and **Page 20**.

On Page 13:

“Further, to test the *in vivo* antimicrobial efficiency of hydrogels, we further evaluated the therapeutic efficacy of HCPF hydrogel in the presence of *S. aureus* colonization. The results showed that HCPF hydrogel could effectively inhibit *S. aureus* in the skin of mice and effectively alleviate AD symptoms under bacterial infection conditions (**Supplementary Fig. S9**). Taken together, the HCPF hydrogels could effectively reduce the inflammation, oxidative damage and *S. aureus* infection in AD tissues and the impairment of the epidermal barrier caused by mechanical scratching to synergistically alleviate AD symptoms.”

On Page 20:

“To verify the *in vivo* antimicrobial efficiency of the hydrogel, we colonized *S. aureus* on the skin surface by applying gauze containing 1×10^6 CFU of *S. aureus* two days before treatment⁷⁴. To determine bacterial numbers in the colonized skin, mice skin was scraped 20 times with cotton swabs to collect the colonized bacteria. Swabs were soaked in PBS for 1 hour, and bacterial suspensions were inoculated in serial dilutions on LB broth agar plates.”

Supplementary Fig. S8 | Evaluation of in vivo antibacterial efficiency of HCPF hydrogels.

a, Timeline of *S. aureus* infected animal experiments. **b**, Representative photographs of the dorsal skin of each group. **c**, Dermatitis score of each group assessed from photographs in **b**. **d**, Epidermal thickness quantified from H&E staining. $n = 3$ mice. In **c** and **d**, $n = 3$ mice. ** and *** indicate $P < 0.01$ and $P < 0.001$ compared by two-tailed Student's t-test, respectively. **e**, Representative H&E staining of skin section. The black boxed area is enlarged below. Scale bar, 100 μm . **f**, Images of survival *S. aureus* bacteria clones. **g**, lg CFU of *S. aureus* of mice skin calculated from **f**. $n = 3$ mice. **** indicates $P < 0.0001$ compared by two-way ANOVA followed by Bonferroni's post hoc test, respectively. All data are shown as mean \pm s.e.m..

Comment #5. In Figure 5 (e and f): The HaCaT monolayer looks immature given that only half of the field of view is covered with cells. Cadherin stability and expression is strongly affected by cell density. How does stretch and stretch + HCF affect E-cadherin expression/localization in confluent monolayers? Images of static controls are needed to better gauge the effect of stretch and hydrogel treatment. How is E-cad deficiency measured?

Response:

(1) According to the reviewer's comments, we have stretched the HaCaT cells with a mature monolayer. We used HC hydrogel without FAKi-lipoLA as the blank control and set static (0% strain) groups. Interestingly, little E-cadherin staining was observed in the unstretched cells. Considering that *in vivo* skin tissues are continuously subjected to stretching at low strain and previous studies have reported that proper mechanical tension promotes intercellular E-cadherin formation, we stretched HaCaT cells at 5% strain for 4 h. The results showed that clear and intact intercellular E-cadherin is formed in the HC and HCF groups. Therefore, we chose cell monolayer with 5% stretch for 4 h to mimic normal skin. Based on this, we further stretched the cells at 30% strain for 4 h. After stretching, E-cadherin in HC group shows a significant deficiency, while this could be rescued in the HCF group. These results demonstrate that FAKi-lipoLA can reduce E-cadherin deficiency in HaCaT cells under large mechanical stretching. We have revised the relevant figure and added related description on **Page 10**.

(2) E-cadherin deficiency is calculated as the ratio of the number of E-Cadherin-uncolored intercellular junctions to the total number of intercellular junctions, *i.e.*, if a cell has intercellular junctions with six surrounding cells and four of these intercellular junctions are not colored by E-cadherin, then the E-cadherin deficiency is 66.66%. We have added a description in **Method** section on **Page 19**.

On **Page 10**:

“E-cadherin is a major component of epidermal intercellular junctions, and its loss of expression correlates with impaired epidermal cell function and morphology⁴⁶. Therefore, we examined the integrity of intercellular E-cadherin by immunofluorescence staining. Interestingly, there is little E-cadherin staining observed in the unstretched cells. Considering that *in vivo* skin tissues are continuously subjected to stretching at low strain and previous studies have reported that proper mechanical tension promotes intercellular E-cadherin formation^{47,48}, we stretched HaCaT cells at 5% strain for 4 h. The results showed that clear and intact intercellular E-cadherin is formed in the HC and HCF groups. Therefore, we used cells with 5% stretching for 4 h to mimic normal skin. Based on this, we further stretched the cells at 30% strain for 4 h, which has been proven to cause mechanical damage to cells⁴⁹. After stretching, E-cadherin in HC group shows a significant deficiency, while this could be rescued in the HCF group (**Fig. 5d,e**).”

On **Page 19**:

“E-cadherin deficiency is calculated as shown below:

$$E - cadherin\ deficiency = \frac{\text{Number of E-cadherin uncolored junctions}}{\text{Total number of intercellular junctions}} \times 100\%.”$$

Figure 5 | Inhibition of FAK protects cells from mechanical damage *in vitro*.

d, Fluorescent staining of HaCaT cells after different stretching with E-cadherin (green) and DAPI (blue). Scale bar, 50 μm . **e**, E-cadherin deficiency of HaCaT cells calculated from staining images in **d**. $n = 4$.

Comment #6. Although the authors mention n values for each experiment, it is unclear whether 'n' stands for the number of technical replicates (i.e. duplicates or triplicates in one experiment) or biological replicates (i.e. different experiments). Given that standard deviation (SD) is reported rather than standard error of the mean (SEM), I presume that the in vitro experiments were performed once with n samples. The general consensus is to report data from at least three experimental repeats.

Response: We thank the reviewer for pointing out this. We are sorry for using the confusing description. In this study, the material characterization of the hydrogels and nanoparticles was performed by replicating test of multiple samples, while the cellular experiments were replicated over multiple experiments. To clarify this, we have changed the relevant images and added a description in the **Statistics** section on **Page 22**.

On **Page 22**:

“Materials experiments were performed by replicating test of multiple samples, and data are shown as mean \pm s.d.. Cell experiments were replicated over multiple experiments, and data are shown as mean \pm s.e.m..”

Comment #7. For in vivo experiments, each mouse can be considered as a separate experiment, which is why SEM rather than SD should be reported. Similarly, quantification of histological images should mention the number of sections/ROIs from how many mice that were used for the analysis.

Response: We thank the reviewer for pointing out this. We have changed the relevant images and data and added a description in the **Statistics** section on **Page 22**.

On **Page 22**:

“For *in vivo* experiments, each slice used for quantification was obtained from different

mice, and data are shown as mean \pm s.e.m..”

Comment #8. The histology in figure 1, 6 and 7 is well executed, but the ‘high magnification’ images are too low in magnification to provide any additional informative compared to the low magnification images. The high magnification images should at least be two-fold higher or more in magnification as to better show where the signal for pFAK and filaggrin is at the cellular level. Also, arrows indicating positive signal will be helpful to the readers who aren’t familiar with DAB staining. Lastly, IGG controls for IF and IHC staining should be included, and the methods of quantification of the staining should be clarified.

Response: We thank the reviewer for pointing out this. Following the reviewer’s suggestion, we have changed the relevant images and added the IGG controls images in **Supplementary Figure S2**.

Figure 2 | pFAK is upregulated in AD skin, especially after scratching.

c, Immunohistochemistry staining of pFAK in each group. Scale bar, 100 μ m for the images above and 25 μ m for the magnified images. The red arrows indicate the area of high DBA staining (high pFAK expression).

Figure 7 | HCPF hydrogels reduce inflammation and epidermal barrier damage of AD mice.

h, Immunohistochemistry staining of filaggrin in each group. Scale bar, 100 μm for the images above and 25 μm for the magnified images. The red arrows indicate the area of high DBA staining (high filaggrin expression).

Supplementary Fig. S7 | Assessment of FAK phosphorylation levels in tissues.

a, Immunohistochemistry staining of pFAK in each group. Scale bar, 100 μm for the images above and 25 μm for the magnified images. The red arrows indicate the area of high DBA staining (high pFAK expression).

Supplementary Figure S2 | Evaluation of inflammation and epidermal barrier damage after scratching AD skins.

I, Representative images of IHC and IF staining of IgG. Scale bar, 50 μm .

Minor Comment

Comment #9. Can the authors elaborate on the self-healing properties of the hydrogel and include a few sentences on the mechanism of 'self-healing' works?

Response: We thank the reviewer for pointing out this. Following the reviewer’s suggestion, we have added the description on **Page 7**.

On **Page 7**:

“To evaluate the self-healing efficiency, we performed time-related strain sweep tests by applying a large shear strain (300%) mimicking mechanical scratching to destroy the HCPF hydrogels. The hydrogel was disrupted at 300% strain (the storage modulus G' decreases significantly to less than the loss modulus G''). When the strain is recovered to 1%, G' recovers rapidly within 20 s, indicating that the disrupted structure is restored (**Fig. 3g**). Adhesion experiments on the skin surface also showed that the dissected HCPF hydrogels could quickly heal within 10 s and withstand the large deformation at the joint (**Fig. 3h**). The good self-healing ability of HCPF hydrogels benefits from the fast dissociation-reconstruction kinetics of boronate ester bonds, *i.e.* the boronate ester bonds dissociate under large strain and undergo rapid addition reactions after the load is removed⁴⁴.”

Comment #10. A red line is used to demarcate the deficient area from the intact area, which is difficult to see for any reader whose color vision is impaired. A white line would be a better choice.

Response: We thank the reviewers for the suggestion. We have replaced the red lines with white lines. In addition, for a clear view of the E-cadherin deficiency, we have magnified the images and indicated the intact E-cadherin and the defective E-cadherin with different arrows.

Figure 7 | HCPF hydrogels reduce inflammation and epidermal barrier damage of

AD mice.

f, Fluorescent staining of E-cadherin (green) and DAPI (blue) in each group. Scale bar, 20 μ m. The white lines denote the deficient area of E-cadherin in the epidermis. In the magnified images, the yellow arrows denote intact intercellular junctions represented by intact E-cadherin, while the white arrows indicate defective E-cadherin.

Comment #11. Line 76: What is mechanosensitization? I'm not sure if this word exists.

Response: We thank the reviewers for pointing out this. 13,800 results can be retrieved in Google Scholar using “mechanosensitization” as the keyword. In some studies, it is also written as “mechanical sensitization”. Mechanosensitization refers to the hypersensitivity of tissues to mechanical stimuli due to inflammation and has been widely used to describe nociceptors on skin or dental nerves (*Journal of Cell Science*, 2018, 131(5): jcs210393.; *Osteoarthritis and cartilage* 27.11 (2019): 1608-1617.; *Nature communications* 11.1 (2020): 2997.).

Comment #12. Typo's that need to be corrected: Line 17: synergistally Line 136 – “Lauri Acid” mistyped? Fig 6a – “Tap Scractch” mistyped? Throughout the paper, ‘tap tears’ should be ‘tape tears’.

Response: We thank the reviewer for pointing out this. We have changed the typo in the manuscript.

Response to Reviewer #3

The manuscript by Jia et al, details the development of a hydrogel dressing for the potential use with atopic dermatitis. Before the manuscript can be considered for publication there are a number of items that should be addressed.

Response: We thank the reviewer for the comment.

Comment #1. Line 45: 'Mechanical scratch is an important cause leading to ...' how this sentence is worded does not make sense, please reword.

Response: We thank the reviewer for pointing out this. As suggested by the reviewer, we have reorganized the words on **Page 3**.

On Page 3:

“Mechanical scratching is an important reason for the aggravation and persistence of AD skin inflammation.”

Comment #2. Line 84: the statement regarding hydrogels and being used for their good biocompatibility. Using the term biocompatibility in general like this is incorrect. Biocompatible refers to the use of a certain material within a specific application. The general term of hydrogel does not specify the material used, not all hydrogels are biocompatible when used as in a skin dressing application.

Response: We thank the reviewer for pointing out this. As suggested by the reviewer, we have deleted the relevant descriptions on **Page 4**.

On Page 4:

“Hydrogels have been considered as promising options for skin dressings due to their drug release ability and diverse customized functions.”

Comment #3. Line 110: refers to patients, this gives the impression that experiments were carried out in humans rather than mice. Please amend.

Response: We thank the reviewer for pointing out this. In **Figure 2a-d** we used both human samples and mouse samples. According to the reviewer's suggestion, we have added a description in the main text on **Page 5**.

On **Page 5**:

“To assess the relationship between FAK phosphorylation and AD, we performed hematoxylin-eosin (HE) staining of skin sections from human AD patients.”

Comment #4. Line 115: refers to rat scratching, though in vivo experiments were carried out in mice Line 119: refers to mast cell infiltration, through these sections were not stained to evaluate mast cell evaluation, please modify statement Line 174: states that the results show the hydrogels can be removed painlessly, though results shown do not demonstrate level of pain felt when hydrogels are removed from skin Line 272: refers to the expression of CCL-20 and TSLP (Fig 5d) though Fig 5d shows the expression of TNF- α and TSLP, please amend text.

Response: We thank the reviewer for pointing out this.

- (1) We have changed the description of “rat” to “mouse”.
- (2) We have added images of mast cells staining in **Supplementary Figure S2**.

Supplementary Figure S2 | Evaluation of inflammation and epidermal barrier damage after scratching AD skins.

a, Representative toluidine blue staining of skin section. The red triangle denotes

dermal mast cells. Scale bar, 100 μ m. n = 5 mice. **b**, Measurement of the density of mast cells for each group after treatments.

(3) We have deleted the statement in Line 174 that hydrogel can be removed painlessly because no pain assessment was performed.

On **Page 7**:

“Tests on human skin also showed that the HCPF hydrogels can adhere to tissue surfaces and withstand various skin deformation or joint movement and can be removed without residue.”

(4) We added the experimental data of CCL-20 in **Figure 5d** and have made corresponding changes in the main text on **Page 10**.

On **Page 10**:

“For HC group, we observed that TNF- α , TSLP and CCL-20 secretions are significantly increased after 30% stretching, while treated with HCF extract significantly reduces their expressions under 30% stretching. Besides, for the 5% stretching groups, there is no difference between the HC and HCF groups, indicating that FAK-lipoLA does not affect the expression of inflammatory factors in the unstretched state (**Fig. 5e**)”

Figure 5 | Inhibition of FAK protects cells from mechanical damage *in vitro*.

f, TNF- α , TSLP and CCL-20 level in the culture medium of HaCaT cells after stretch. n = 3.

Comment #5. The text that is referring to Fig 6 is incorrect, example, line 305/306

refers to mast cell infiltration by toluidine blue staining (Fig 6e,f) though Fig 6f shows epidermal thickness.

Response: We thank the reviewer for pointing out this. We have corrected the mislabeling.

Comment #6. Line 337: text states (Fig h,i) please include figure number.

Response: We thank the reviewer for pointing out this. This text state indicates (**Fig.7 h,i**). We have corrected the mislabeling.

REVIEWERS' COMMENTS

Reviewer #1 (Remarks to the Author):

The authors have addressed the comments properly. I would like to suggest the manuscript for publication.

Reviewer #2 (Remarks to the Author):

The authors have satisfactorily addressed my comments. I did, however, find two additional minor errors that should be addressed.

Line 314: "Besides, for the 5% stretching groups, there is no difference between the HC and HCF groups, indicating that FAK-LipoLA does not affect the expression of inflammatory factors in the unstretched state (Fig. 5f)."

This statement is incorrect as the cells are stretched by 5%.

The text in line 478 refers to rats, but the experiments were conducted in mice.

Response Letter

Re: # NCOMMS-22-45011A

Title: A Multifunctional Hydrogel Dressing Integrating FAK Inhibition and ROS Scavenging for Mechano-chemically Synergistic Treatment of Atopic Dermatitis

Dear Reviewers,

We would like to thank you for your time to review our manuscript and your invaluable and constructive comments. We have revised the manuscript according to the comments and significantly improved the quality of our paper. For convenience, the reviewers' comments have been marked in our current response letter *in nattier blue and italic font* and the response to the comment is in black. Changes in the manuscript are marked in **red**.

Response to Reviewer #1

The authors have addressed the comments properly. I would like to suggest the manuscript for publication.

Response: We thank the reviewer for the encouraging comment.

Response to Reviewer #2

The authors have satisfactorily addressed my comments. I did, however, find two additional minor errors that should be addressed.

Response: We thank the reviewer for the encouraging comment.

Line 314: “Besides, for the 5% stretching groups, there is no difference between the HC and HCF groups, indicating that FAK-LipoLA does not affect the expression of inflammatory factors in the unstretched state (Fig. 5f).” This statement is incorrect as the cells are stretched by 5%.

Response: We thank the reviewer for pointing out this, we have fixed the error in the description.

In line 314:

“Besides, for the 5% stretching groups, there is no difference between the HC and HCF groups, indicating that FAK-lipoLA does not affect the expression of inflammatory factors under 5% stretching (**Fig. 5f**).”

The text in line 478 refers to rats, but the experiments were conducted in mice.

Response: We thank the reviewer for pointing out this, we have fixed the error in the description.

In line 477:

“In our AD model, tape tearing was used to simulate steady mechanical stimulation while the lesioned skin was covered with gauze to avoid spontaneous scratching by the mice.”